# Hyperactivity of indirect pathway-projecting spiny projection neurons promotes compulsive behavior

Sean C. Piantadosi[1,2,3,7], Elizabeth E. Manning [2,4,7], Brittany L. Chamberlain [1,2,7], James Hyde[2,5], Zoe LaPalombara [1,2], Nicholas M. Bannon[2], Jamie L. Pierson[2], Vijay M. K Namboodiri [6] & Susanne E. Ahmari [1,2] ✉

Compulsive behaviors are a hallmark symptom of obsessive compulsive disorder (OCD). Striatal hyperactivity has been linked to compulsive behavior generation in correlative studies in humans and causal studies in rodents. However, the contribution of the two distinct striatal output populations to the generation and treatment of compulsive behavior is unknown. These populations of direct and indirect pathway-projecting spiny projection neurons (SPNs) have classically been thought to promote or suppress actions, respectively, leading to a long-held hypothesis that increased output of direct relative to indirect pathway promotes compulsive behavior. Contrary to this hypothesis, here we find that indirect pathway hyperactivity is associated with compulsive grooming in the *Sapap3*-knockout mouse model of OCD-relevant behavior. Furthermore, we show that suppression of indirect pathway activity using optogenetics or treatment with the first-line OCD pharmacotherapy fluoxetine is associated with reduced grooming in *Sapap3*-knockouts. Together, these findings highlight the striatal indirect pathway as a potential treatment target for compulsive behavior.

Convergent evidence from structural and functional imaging studies has broadly linked obsessive compulsive disorder (OCD) symptoms to abnormalities in the striatum, specifically highlighting hyperactivity in the caudate at baseline and during symptom provocation[1–8]. Effective treatment with serotonin reuptake inhibitors (SRIs) normalizes striatal hyperactivity, suggesting that this abnormal activity underlies OCD symptoms[9,10]. Complementary findings from rodents provide a causal link between hyperactivity in striatal subregions and generation of compulsive behavior relevant to symptoms of OCD[11–14]. Together, these data strongly implicate modulation of striatal activity in the generation and treatment of

compulsive behavior. However, they do not provide insight into how distinct classes of striatal spiny projection neurons (SPNs), which can be defined by their dopamine receptor expression patterns or projection targets in the basal ganglia, contribute to the pathophysiology of OCD. Classically, direct pathway-projecting SPNs (dSPNs) that preferentially express the dopamine D1-receptor are thought to promote movement, while indirect pathway-projecting SPNs (iSPNs) that preferentially express the dopamine D2-receptor are thought to suppress movement[15–17] (though see[18–22]). Understanding how activity of these specific striatal neuronal populations and their associated pathways through the basal ganglia contribute to the generation of

¹Center for Neuroscience, University of Pittsburgh, Pittsburgh, PA, USA. ²Department of Psychiatry, University of Pittsburgh, Pittsburgh, PA, USA. ³Department of Anesthesiology and Pain Medicine, University of Washington, Seattle, WA, USA. ⁴School of Biomedical Sciences and Pharmacy, University of Newcastle, Callaghan, NSW, Australia. ⁵Department of Biology, Southern Arkansas University, Magnolia, AK, USA. ⁶Department of Neurology, University of San Francisco, San Francisco, CA, USA. ⁷These authors contributed equally: Sean C. Piantadosi, Elizabeth E. Manning, Brittany L. Chamberlain. ✉e-mail: ahmarise@upmc.edu

compulsive behaviors may help guide the development of new treatment strategies.

A longstanding theory suggests that OCD symptoms result from increased dSPN activity and/or decreased iSPN activity[5,8,23]. While evidence for this idea has been observed in other models of pathological repetitive behaviors[24–29], limited support exists in the context of intrinsically generated compulsive behaviors relevant to OCD[30,31], with no in vivo assessments of activity balance between pathways conducted to date. Here we tested this theory by directly examining activity of SPN subtypes in freely moving Sapap3-knockout (Sapap3-KO) mice, a strain that exhibits SRI-sensitive, OCD-relevant behavioral phenotypes including compulsive grooming behavior, along with well-established structural and physiological striatal abnormalities[12,31–38].

Using in vivo calcium imaging in Sapap3-KOs, we found that hyperactivity in the central striatum (CS), which has previously been demonstrated to be critical for compulsive grooming in this model[12], results from an increased number of activated neurons at the onset of grooming bouts. We then used genetic and circuit-based strategies to isolate putative direct and indirect pathway-projecting SPNs. Surprisingly, we found that hyperactivity in the indirect pathway is associated with excessive grooming, and that a reduction of indirect pathway hyperactivity is associated with decreased grooming. Together, these results point to an unexpected role for iSPNs in the generation of OCD-relevant compulsive behaviors and treatment response.

## Results

### Central striatum is hyperactive during grooming in Sapap3-KO mice

To investigate the net output of all CS neuron subtypes during compulsive grooming behavior, we expressed the genetically encoded calcium indicator GCaMP6m via human synapsin promoter and implanted a gradient refractive index (GRIN) lens above the injection site (Fig. 1a; Fig.S1). After 5 weeks virus incubation, grooming behavior was assessed and central striatal calcium activity was recorded using a single-photon miniature microscope during a 40 min session. Consistent with previous reports[12,33,34], Sapap3-KO mice spent significantly more time grooming and engaged in significantly more grooming bouts than wildtype (WT) littermates (Fig. 1c; Fig.S2a). This increase in the number of grooming bouts was observed across all grooming subtypes (Fig. S2a, b). No change in grooming bout duration was observed between Sapap3-KOs and -WTs (Fig. S2c, d). Calcium activity of hundreds of putative central striatal neurons was segmented as previously described (Fig. 1d; Supplementary Movie 1)[39,40], and a continuous fluorescence time series was extracted for each neuron (see Methods). We first evaluated the degree to which central striatal population activity was able to decode spontaneous grooming displayed by both Sapap3-WT and -KO mice. Using a binary RUSBoost classifier, a linear classifier that minimizes the effect of class imbalance[41], we found that central striatal population activity is better able to classify grooming behavior compared to shuffled grooming behavior in both genotypes (Fig. 1e, left; Supplementary Data 1). Furthermore, we observed that grooming behavior was more accurately classified in Sapap3-KOs compared to Sapap3-WTs (Fig. 1e, right) (as determined by F1 score, the harmonic mean of precision and recall), indicating that abnormal central striatal activity may contribute to generation of excessive grooming.

We next examined when and how central striatal activity differed during excessive grooming in KOs versus normal grooming behavior in WTs. CS activity increased transiently at the onset of grooming behavior significantly more in Sapap3-KOs compared to -WTs, and this increase persisted throughout grooming bouts (Fig. 1f). While no change in calcium fluorescence was observed between the pre-grooming and grooming period in -WT mice, a significant increase was observed in Sapap3-KOs (Fig. 1g). This grooming-onset increase in fluorescence was observed when face and body grooming were

analyzed separately (Fig. S2e, f) or together (Fig. S2g), while no difference was observed for hind leg scratching (Fig. S2h). Together, these data indicate that increased CS activity is not associated with excessive hind leg scratching in Sapap3-KOs. Additionally, while Sapap3-KOs engaged in fewer locomotion bouts relative to -WTs (Fig. S2i), no difference in locomotion-onset fluorescence was observed (Fig. S2j). Furthermore, the grooming-onset-associated increase in fluorescence in central striatal neurons was also associated with a significantly elevated calcium event rate in Sapap3-KOs compared to Sapap3-WTs during grooming (Fig. 1h; left). In contrast, no difference was observed in the overall calcium event rate during all non-grooming periods (Fig. 1h; right), indicating that CS hyperactivity in KOs is restricted to the time of engagement in excessive grooming behavior. The enhanced predictive strength and CS hyperactivity in Sapap3-KOs could result from a difference in the number of cells active during grooming. Using a statistical method to classify neurons as activated or inhibited at grooming onset (a period −0.5 s before to 3 s after grooming start)[42], we found that all mice in both genotypes had central striatal neurons with grooming-onset modulated activity (Fig. 1i). Compared to -WTs, Sapap3-KOs had a significantly greater percentage of grooming-onset activated neurons (Fig. 1j, k). The increase in activated neurons at grooming onset in -KOs was robust to changes in the standard deviation threshold used for classification (Fig. S2k) and was not due solely to the greater number of grooming bouts engaged in by -KOs, since restricting the number of bouts analyzed in Sapap3-KOs to the mean bout number in -WT mice still produced a significant increase in activated neurons with a small reduction in effect size due to reduced bout number (Fig. S2l). No differences were observed in percentage of grooming-onset inhibited neurons (Fig. 1k). These data suggest compulsive behavior may result from central striatal hyperactivity driven by increased recruitment of activated SPNs, which in turn may yield stronger population decoding of grooming behavior in KOs.

### D1-SPNs are not hyperactive at onset of compulsive behavior

Given that pharmacological activation of dopamine D1-receptors preferentially expressed by direct pathway-projecting SPNs produces grooming[43–46] and direct pathway hyperactivity has long been hypothesized to underlie compulsive behavior[4,6,8,23], we predicted that increased D1-SPN activity at grooming onset would explain the central striatal hyperactivity observed in Sapap3-KOs (Fig. 1). To selectively image D1-SPNs, we injected Cre-dependent DIO-GCaMP6m into CS of double transgenic heterozygous D1-Cre+/−/Sapap3-KO and D1-Cre+/−/Sapap3-WT littermate controls and implanted GRIN lenses (Fig. 2a; Fig. S3). Following viral expression, mice underwent a 40 min grooming and imaging session (Fig. 2b; Supplementary Movie 2). As before, Sapap3-KOs spent significantly more time grooming and engaged in significantly more grooming bouts than Sapap3-WTs (Fig. 2c). Surprisingly, we found no significant elevation of D1-SPN calcium fluorescence in Sapap3-KOs compared to -WTs at grooming onset (Fig. 2d). Similarly, no genotype differences were detected in D1-SPN calcium event rates during either grooming or non-grooming epochs (Fig. 2e). Furthermore, WT and KO mice showed no differences in percentage of D1-SPNs that were activated or inhibited at grooming onset (Fig. 2f) or in the accuracy of grooming classification (as assessed by F1 score) by D1-SPN activity (Fig. 2g, h). Contrary to our hypothesis, these data suggest that increased CS activity at grooming onset in KOs is not explained by activity of D1-SPNs.

While we did not detect grooming-onset activity differences between Sapap3-KOs and -WTs in the combined D1-SPN population average, we noted that individual neurons displayed substantial heterogeneity in activity profiles (Fig. 2d). Because of prior evidence that different activity patterns of individual striatal neurons correspond to specific phases of grooming[47,48], we next asked if specific subpopulations of D1-SPNs distinguish grooming in Sapap3-KOs vs -WTs. We

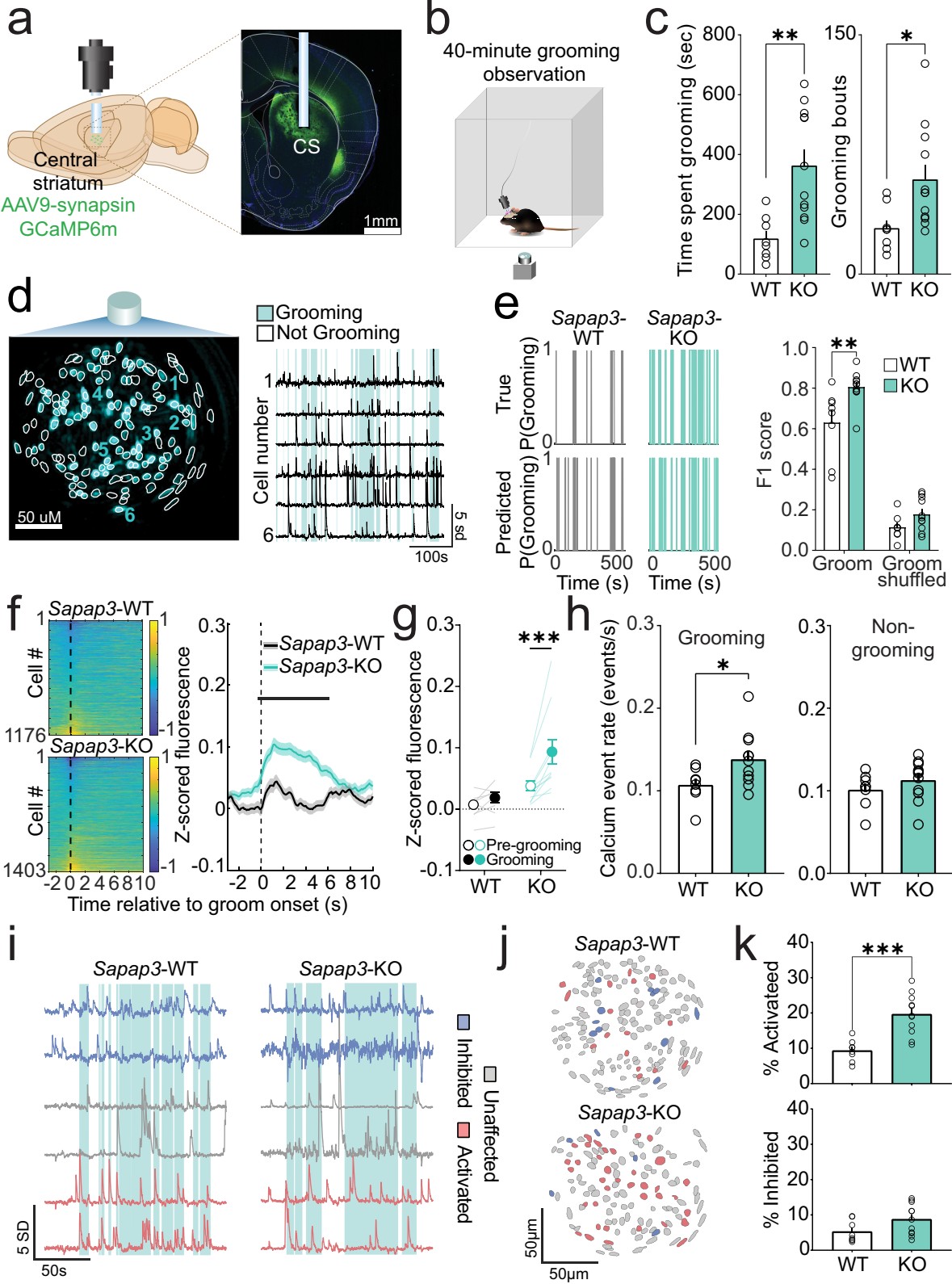

used an unsupervised spectral clustering algorithm[49] to group neurons with similar temporal dynamics in their average grooming-onset aligned activity. This approach identified 8 functional subpopulations (a.k.a. clusters) of grooming-related activity (Fig. 2i; Fig. S4). Neurons were identified in each of these 8 clusters in each mouse of both genotypes. Clusters were generally separable in high-dimensional principal component space from every other cluster (Fig. S4b, c),

supporting the idea that functionally distinct groups of D1-SPNs may differentially control distinct grooming sequence phases in both *Sapap3*-WT and -KO mice. Several clusters displayed increased activity (above baseline) before (Cluster 2) and at different phases throughout (Clusters 1, 3, 4, 6) grooming behavior in both genotypes. Other clusters displayed reductions in activity during grooming (Cluster 5) or limited grooming-associated changes (Clusters 7 and 8). Comparing

**Fig. 1 | Central striatum is hyperactive in *Sapap3*-KO mice during grooming.**
**a** Experimental design for imaging central striatal (CS) neurons (left) and representative histological image of GCaMP6m expression and lens placement in CS (right). Brain atlas overlay used with permission of Elsevier Science and Technology Journals from Paxinos and Franklin's the Mouse Brain in Stereotaxic Coordinates, Franklin Keith B.J., Paxinos, George, volume 5, copyright year 2019; permission conveyed through Copyright Clearance Center, Inc. **b** Schematic of behavioral apparatus. **c** *Sapap3*-KO mice (*n* = 11, 6 male / 5 female) spend significantly more time grooming (left; two-tailed unpaired *t*-test, *t*(18) = 3.51, *p* = 0.003) and engage in significantly more grooming bouts (right; two-tailed unpaired, *t*-test *t*(18) = 2.56, *p* = 0.02) than *Sapap3*-WT mice (*n* = 8, 5 male/3 female). **d** Contour map of putative CS neurons overlaid on peak-to-noise ratio image (left). Individual traces of selected neurons in field of view (FOV) overlaid on top of grooming (blue) behavior (right). **e** Probability of grooming in a representative *Sapap3*-WT (left, *n* = 8, 5 male / 3 female) and *Sapap3*-KO mouse (right, *n* = 11, 6 male/5 female). Top traces represent real grooming behavior; bottom represents grooming predicted via RUSBoost decoder. SPN population F1 score 2 x (precision x recall) / (precision + recall) is greater for grooming than for shuffled grooming behavior [two-way repeated measures ANOVA; main effect of behavior type; *F*(1,17) = 81.41, *p* < 0.0001]. Main effect of genotype [*F*(1,17) = 3.57, *p* = 0.001]. Interaction between genotype and behavior type [*F*(1,17) = 0.78, *p* = 0.04]. Grooming F1 score was improved in *Sapap3*-KOs (teal) relative to -WTs (white) [Sidak's multiple comparisons test, *t*(34) = 3.69, *p* = 0.002]. **f** Trial averaged activity aligned to grooming start across all cells in *Sapap3*-WT (top; *n* = 1176 neurons) and *Sapap3*-KO mice (bottom; *n* = 1403 neurons). Black dotted line indicates groom start. CS neurons display elevated activity at grooming start in *Sapap3*-KOs compared to *Sapap3*-WTs (two-tailed unpaired *t*-test, all significant *t*(2577) ≥ 3.61, *p* ≤ 0.00038; thick black line in figure). Shading indicates ±SEM. **g** Mean Z-score fluorescence during pre-grooming (open circles) and grooming (closed circles) periods for *Sapap3*-KO (teal, *n* = 11, 6 male/5 female) and -WT (black, *n* = 8, 5 male / 3 female) mice. Significant main effects of time [Two-way repeated measures ANOVA; *F*(1,17) = 16.43, *p* = 0.0008] and genotype [*F*(1,17) = 8.97, *p* = 0.008] and an interaction between time and genotype [*F*(1,17) = 7.16, *p* = 0.016] were detected. No difference between pre-grooming and grooming mean fluorescence was detected for WT mice (*p* = 0.61), while a significant increase in grooming fluorescence relative to pre-grooming was observed for KOs [*t*(17) = 5.19, *p* = 0.0001]. **h** Calcium event rates are elevated during compulsive grooming in *Sapap3*-KOs (*n* = 11, 6 male/5 female) relative to -WT (*n* = 8, 5 male/3 female) mice (two-tailed unpaired *t*-test, *t*(17) = 2.31, *p* = 0.033). No difference in calcium event rates for all other times during the session (two-tailed unpaired *t*-test, *t*(17) = 1.07, *p* = 0.30). **i** Representative traces of cells classified as grooming-onset activated (red), grooming-onset inhibited (blue), and unaffected at grooming onset (grey) in a *Sapap3*-WT (left) and *Sapap3*-KO mouse (right). **j** Representative contour maps of individual CS neurons colored according to (**i**). **k** *Sapap3*-KO mice (*n* = 11, 6 male/5 female) have a significantly greater percentage of CS neurons activated at grooming onset compared to *Sapap3*-WT mice (*n* = 8, 5 male/3 female, two-tailed unpaired *t*-test, *t*(17) = 4.56, *p* = 0.0003). No difference in percentage of grooming-onset inhibited cells between genotypes (*p* = 0.08). \*\*\**p* ≤ 0.001, \*\**p* ≤ 0.01, \**p* ≤ 0.05. Data are presented as mean values +/− SEM. Source data are provided as a Source Data file.

the spectral clusters to our activity-based grooming-onset classification (Fig. 2f), we find that D1-SPNs identified as grooming-start-activated overlapped substantially with spectral clusters typified by increased activity before or just after grooming onset (Fig. S5a, b; Clusters 1, 2, & 3 comprise a total of 86% of grooming-onset activated cells for WTs and KOs combined), with much less representation in other clusters (14% of cells in Clusters 4-8). In addition, grooming-start-inhibited neurons had very little overlap with clusters showing grooming onset activity (Fig. S5a, b). These data suggest that activity of D1-SPNs during grooming is similarly distributed across functional clusters in WT and KO mice.

While D1-SPN average population activity at grooming onset was not different in *Sapap3*-WT vs. -KO mice (Fig. 2d–f), it is possible that differential recruitment of D1-SPNs within these functional clusters is associated with elevated grooming. We found that *Sapap3*-KOs had lower proportions of D1-SPNs that were active prior to grooming onset (Cluster 2) and toward the end of a grooming sequence (Cluster 6; median grooming bout duration: KO = 7.5 s, WT = 4.4 s [Mann-Whitney test, *U* = 10, *p* = 0.08]). Relative to -WT mice, *Sapap3*-KOs also had increased proportions of neurons that were activated shortly after grooming onset (Cluster 3) and that were largely unaffected by grooming (Cluster 8). Together these data from functional clusters also did not support our hypothesis that hyperactivity of D1-SPNs at grooming onset is the primary source of overall striatal hyperactivity observed in Fig. 1. Specifically, no change (Cluster 1) or even a reduction (Cluster 2) in the proportion of activated neurons was detected in clusters whose activity increased immediately at grooming onset in KOs. Instead, differences in proportions of WT and KO D1-SPNs were identified in clusters with activity during, rather than preceding, grooming (Cluster 3 and 6).

**D2-SPNs are hyperactive at the onset of compulsive behavior**
Given the surprising finding that D1-SPN activity could not explain striatal hyperactivity associated with grooming onset in *Sapap3*-KOs (Fig. 1), we next investigated whether D2-SPNs might unexpectedly be the primary source of SPN hyperactivity. We generated double transgenic mice by crossing adenosine receptor A2a-Cre (A2a-Cre⁺/⁻) mice with *Sapap3*⁺/⁻ mice. A2a-Cre⁺/⁻/*Sapap3*-KO and -WT littermates were injected with a Cre-dependent genetically encoded calcium indicator (DIO-GCaMP6m) to achieve selective expression in D2-SPNs, and a GRIN lens was implanted into CS (Fig. 3a; Fig. S6a). After viral expression, mice underwent a 40 min grooming observation (Fig. 3b; Supplementary Movie 3). Similar to previous cohorts, *Sapap3*-KOs spent more time grooming and engaged in significantly more grooming bouts than -WTs (Fig. 3c). Strikingly, we found that D2-SPN activity was robustly increased in *Sapap3*-KOs at onset of grooming relative to -WTs (Fig. 3d). D2-SPN calcium event rates during grooming were also significantly increased in *Sapap3*-KOs while no genotype differences were observed in event rates during non-grooming times (Fig. 3e), indicating hyperactivity was restricted to elevated grooming behavior. Much like our imaging findings in all CS neuron cell types (Fig. 1), D2-SPN hyperactivity during excessive grooming in *Sapap3*-KOs was associated with increased recruitment of grooming-onset activated D2-SPNs relative to -WTs (Fig. 3f); this was not a consequence of a difference in the number of grooming bouts across genotypes (Fig. S6b). Interestingly, whereas D1-SPN activity was able to predict grooming equally well in *Sapap3*-KOs and -WTs (Fig. 2h), D2-SPN population activity was able to more accurately predict excessive grooming in -KOs compared to normal -WT grooming behavior as measured by the F1 score (Fig. 3h). Together, these data indicate that D2-SPN hyperactivity is likely the primary source of overall striatal hyperactivity associated with compulsive behavior.

We next determined whether recruitment of functional clusters of D2-SPNs was different during KO versus WT grooming. D2-SPNs were represented in the same 8 clusters identified within the D1-SPN population (Fig. 3i), consistent with prior reports that overall patterns of striatal activity during spontaneous behavior are similar between genetically defined SPN subtypes[18,19]. Similar to what we observed in D1-SPNs (Fig. S5a, b), D2-SPN functional clusters typified by increases in fluorescence around grooming start overlapped substantially with neurons identified as grooming-start-activated (Fig. 3f) irrespective of genotype, while clusters characterized by reductions in activity overlapped with neurons identified as grooming-start-inhibited (Fig. S5c-d). Compared to -WTs, *Sapap3*-KOs had increased proportions of D2-SPNs with activity increases at the onset of grooming (Cluster 1) and activity increases during grooming behavior (Clusters 3 & 4). In contrast, KOs had reduced proportions of functional D2-SPN clusters that displayed limited

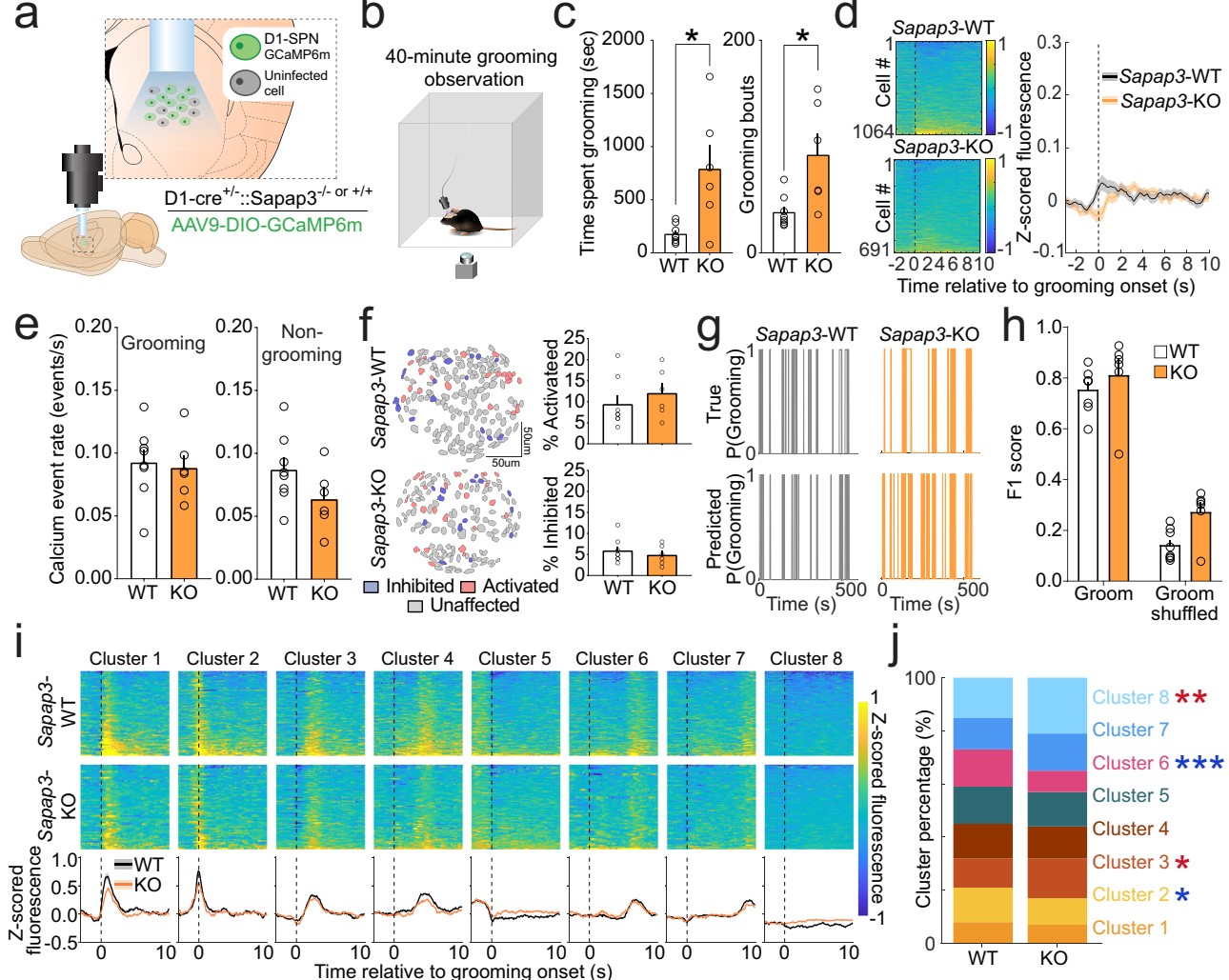

**Fig. 2 | D1-SPNs are not hyperactive at grooming onset in *Sapap3*-KO mice.** **a** Experimental design for selective imaging of D1-SPNs. Brain atlas overlay used with permission of Elsevier Science and Technology Journals from Paxinos and Franklin's the Mouse Brain in Stereotaxic Coordinates, Franklin Keith B.J., Paxinos, George, volume 5, copyright year 2019; permission conveyed through Copyright Clearance Center, Inc. **b** Schematic of behavioral apparatus. **c** *Sapap3*-KO ($n = 6$: 4 male / 2 female) mice spend significantly more time grooming (left; two-tailed Mann-Whitney test, $U = 8$, $p = 0.04$) and engage in significantly more grooming bouts (right, two-tailed Mann-Whitney test, $U = 5$, $p = 0.01$) than *Sapap3*-WT ($n = 8$: 3 male/5 female) mice. **d** Trial averaged activity aligned to grooming start across all cells in *Sapap3*-WT (top; $n = 1064$) and *Sapap3*-KO mice (bottom; $n = 691$). Black dotted line indicates groom start. No significant difference between *Sapap3*-WTs and *Sapap3*-KOs in overall grooming onset D1-SPN activity (two-tailed unpaired t-test, all $t(1753) \leq 2.81$, $p \geq 0.00038$). Shading indicates ±SEM. **e** D1-SPN calcium event rates are not significantly different during grooming (left) and non-grooming (right) periods in *Sapap3*-KOs ($n = 6$: 4 male / 2 female) compared to -WT ($n = 8$: 3 male / 5 female) mice (two-tailed Mann-Whitney test, $p = 0.41$). **f** Contour map of D1-SPNs colored according to their activity (red=activated, blue=inhibited, grey=unaffected) at onset of grooming (left). Average percentage of activated and inhibited D1-SPNs in -WT ($n = 8$: 3 male/5 female) and *Sapap3*-KO ($n = 6$: 4 male/2

female) mice (two-tailed Mann-Whitney test, $p = 0.30$ and $p = 0.56$, respectively). **g** Probability of grooming in a representative *Sapap3*-WT (left) and *Sapap3*-KO mouse (right). Top traces represent real grooming behavior; bottom represents grooming predicted via RUSBoost decoder based on D1-SPN population activity. **h** D1-SPN population decoding F1 score is greater for grooming than for shuffled grooming (Two-way repeated measures ANOVA, main effect of behavior type; $F(1,12) = 566.9$, $p < 0.0001$). No main effect of genotype ($F(1,12) = 3.17$, $p = 0.10$) or interaction between genotype and behavior type was observed ($F(1,12) = 2.25$, $p = 0.16$). **i** Functional clustering of trial averaged grooming-onset activity in D1-SPNs in -WT (top, $n = 8$: 3 male / 5 female) and *Sapap3*-KO mice (middle, $n = 6$: 4 male /2 female) represented as a heatmap. Functional clustering identified 8 distinct functional clusters (bottom). Mean grooming-onset activity of each cluster in -WT (black) and *Sapap3*-KOs (orange). Black dotted line indicates grooming. **j** Compared to -WTs, *Sapap3*-KOs have a greater proportion of Cluster 3 (Chi-square test of proportions, $X^2(1) = 4.77$, $p = 0.03$) and Cluster 8 ($X^2(1) = 8.76$, $p = 0.003$) D1-SPNs; a significantly lower proportion of Cluster 2 ($X^2(1) = 3.62$, $p = 0.05$) and Cluster 6 ($X^2(1) = 16.53$, $p = 0.0001$) D1-SPNs; and no changes in Clusters 1, 4, 5, and 7. ***$p \leq 0.001$, **$p \leq 0.01$, *$p \leq 0.05$. Data are presented as mean values +/− SEM. Source data are provided as a Source Data file.

grooming-associated changes in activity (Clusters 7 and 8). Relatively few D2-SPNs were classified as exhibiting pre-grooming-onset activity (Cluster 2), and no differences in the proportion of SPNs in this cluster were observed between genotypes (Fig. 3j). Overall, these data suggest that hyperactivity in functional clusters of D2-SPNs activated immediately at grooming onset may contribute to the generation of increased grooming behavior in *Sapap3*-KOs.

**Optogenetic inhibition of striatopallidal iSPN activity reduces excessive grooming behavior**

These data highlight that neural activity changes in D2-SPNs are associated with compulsive grooming behavior in *Sapap3*-KOs and may represent an important treatment target. Because D2-receptors are preferentially expressed on indirect pathway-projecting SPNs, circuit-based neurosurgical treatments targeting the indirect pathway

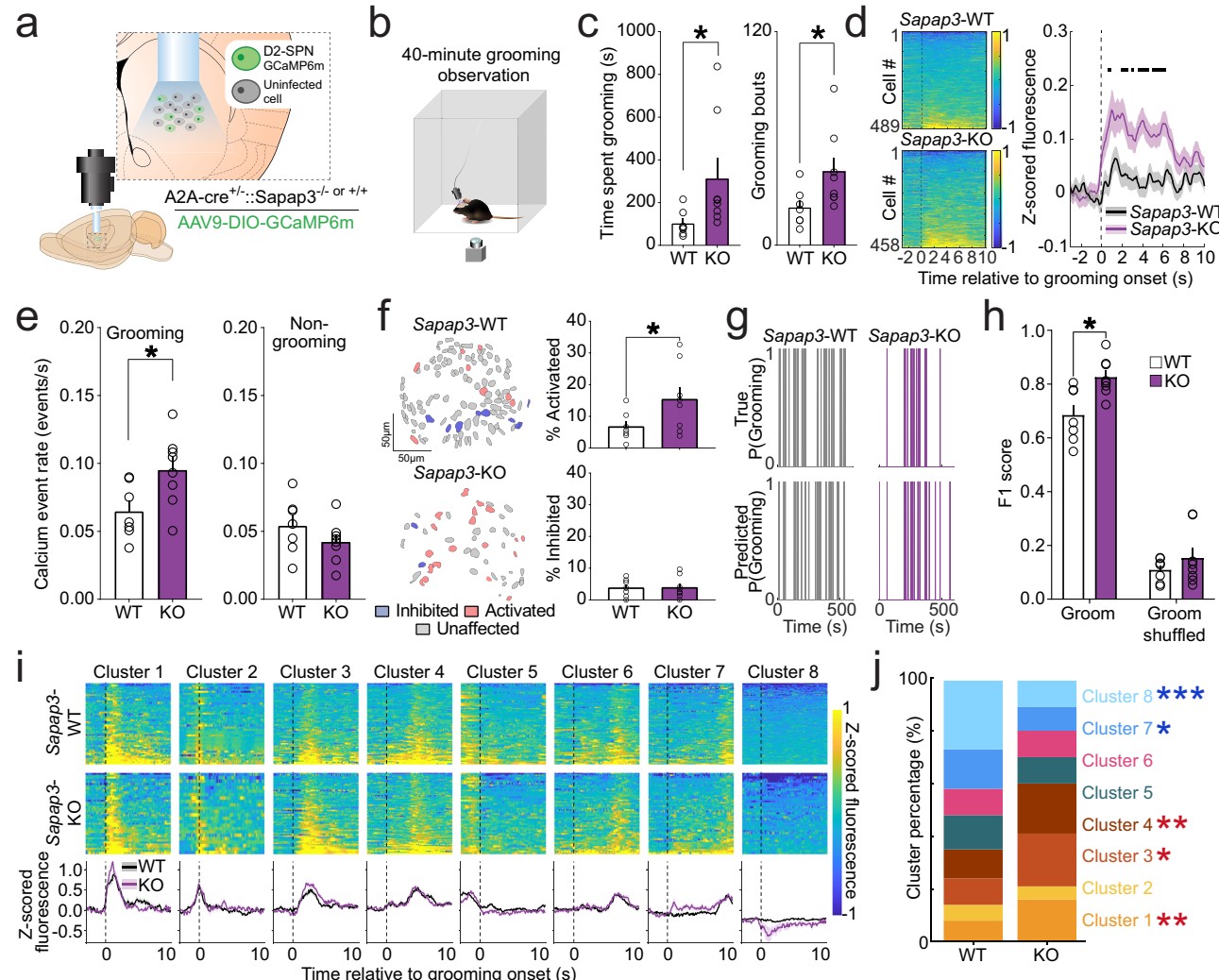

**Fig. 3 | D2-SPNs are hyperactive at grooming onset in *Sapap3*-KO mice.**
**a** Experimental design for selective imaging of D2-SPNs. Brain atlas overlay used with permission of Elsevier Science and Technology Journals from Paxinos and Franklin's the Mouse Brain in Stereotaxic Coordinates, Franklin Keith B.J., Paxinos, George, volume 5, copyright year 2019; permission conveyed through Copyright Clearance Center, Inc. **b** Schematic of behavioral apparatus. **c** *Sapap3*-KO ($n = 8$: 6 male/ 2 female) mice spend more time grooming (left; two-tailed Mann-Whitney test, $U = 8$, $p = 0.02$) and engage in significantly more grooming bouts (right; two-tailed Mann-Whitney test, $U = 9$, $p = 0.03$) than *Sapap3*-WT ($n = 7$: 5 male/ 2 female) mice. **d** Trial averaged activity aligned to grooming start across all cells in *Sapap3*-WT (top; $n = 489$) and *Sapap3*-KO mice (bottom; $n = 458$). Black dotted line indicates groom start. Median grooming bout duration for WTs (4.4 s) and KOs (6.1 s) two-tailed Mann-Whitney test ($U = 11$, $p = 0.06$). Groom-start aligned D2-SPN activity is significantly elevated in *Sapap3*-KOs compared to -WTs (all significant $t(945) \geq \pm 3.58$, $p \leq 0.00038$; thick black lines in figure). Shading indicates ±SEM. **e** D2-SPN calcium event rates are elevated during grooming (left; two-tailed Mann-Whitney test, $U = 7$, $p = 0.01$) and unchanged during non-grooming (right; $p = 0.33$) periods in *Sapap3*-KOs ($n = 8$: 6 male/ 2 female) compared to -WT ($n = 7$: 5 male/ 2 female) mice (**f**) Contour map of D2-SPNs colored according to their activity (red=activated, blue=inhibited, grey=unaffected) at the onset of grooming (left). *Sapap3*-KOs ($n = 8$: 6 male/ 2 female) have a greater proportion of groom-onset activated D2-SPNs than -WT ($n = 7$: 5 male/ 2 female) mice (Mann-Whitney test, $U = 13$, $p = 0.05$), with no changes in the proportion of inhibited neurons ($p = 0.49$).

**g** Probability of grooming in a representative *Sapap3*-WT (left) and *Sapap3*-KO mouse (right). Top traces represent real grooming behavior; bottom represents grooming predicted via an RUSBoost classifier based on D2-SPN population activity. **h** D2-SPN population decoding F1 score is greater for grooming than for shuffled grooming [Two-way repeated measures ANOVA, main effect of behavior type; $F(1,13) = 826.0$, $p < 0.0001$]. A main effect of genotype [$F(1,13) = 6.04$, $p = 0.03$] and significant interaction between genotype and behavior type was observed [($F(1,13) = 4.98$, $p = 0.04$]. Classifier accuracy was higher during excessive grooming in *Sapap3*-KOs ($n = 8$: 6 male/ 2 female) compared to -WT ($n = 7$: 5 male/ 2 female) mice [Sidak's multiple comparisons test, $t(26) = 3.24$, $p = 0.007$], with no difference when the classifier was trained on shuffled data ($p = 0.54$). **i** Functional clustering of trial averaged grooming-onset activity in D2-SPNs in -WT (top) and *Sapap3*-KO mice (middle) represented as a heatmap. Functional clustering identified 8 distinct functional clusters (bottom). Black dotted line indicates groom start. Mean grooming-onset activity of each cluster in -WT (black) and *Sapap3*-KO mice (purple). **j** Compared to -WTs, *Sapap3*-KO have a greater proportion of Cluster 1 (Chi-square test of proportions, $X^2(1) = 6.75$, $p = 0.01$), Cluster 3 ($X^2(1) = 5.076$, $p = 0.024$) and Cluster 4 ($X^2(1) = 7.65$, $p = 0.006$) D2-SPNs; a significantly lower proportion of Cluster 7 ($X^2(1) = 5.52$, $p = 0.02$) and Cluster 8 ($X^2(1) = 27.09$, $p < 0.0001$) D2-SPNs; and no changes in Clusters 2, 5, and 6. ***$p \leq 0.001$, **$p \leq 0.01$, *$p \leq 0.05$. Data are presented as mean values +/- SEM. Source data are provided as a Source Data file.

could be an effective treatment strategy for compulsive behavior disorders like OCD[50,51]. To evaluate this possibility, we used a circuit-specific strategy to selectively manipulate striatopallidal iSPNs projecting to the external segment of the globus pallidus (GPe). To compare this approach to our prior genetic targeting of D2-SPNs, we injected a retrograde AAV2-Cre-eGFP into GPe of WT mice ($n = 4$) and 10 weeks later performed in situ hybridization to examine overlap between retrogradely expressed eGFP and markers for D1- (*Drd1a*) and

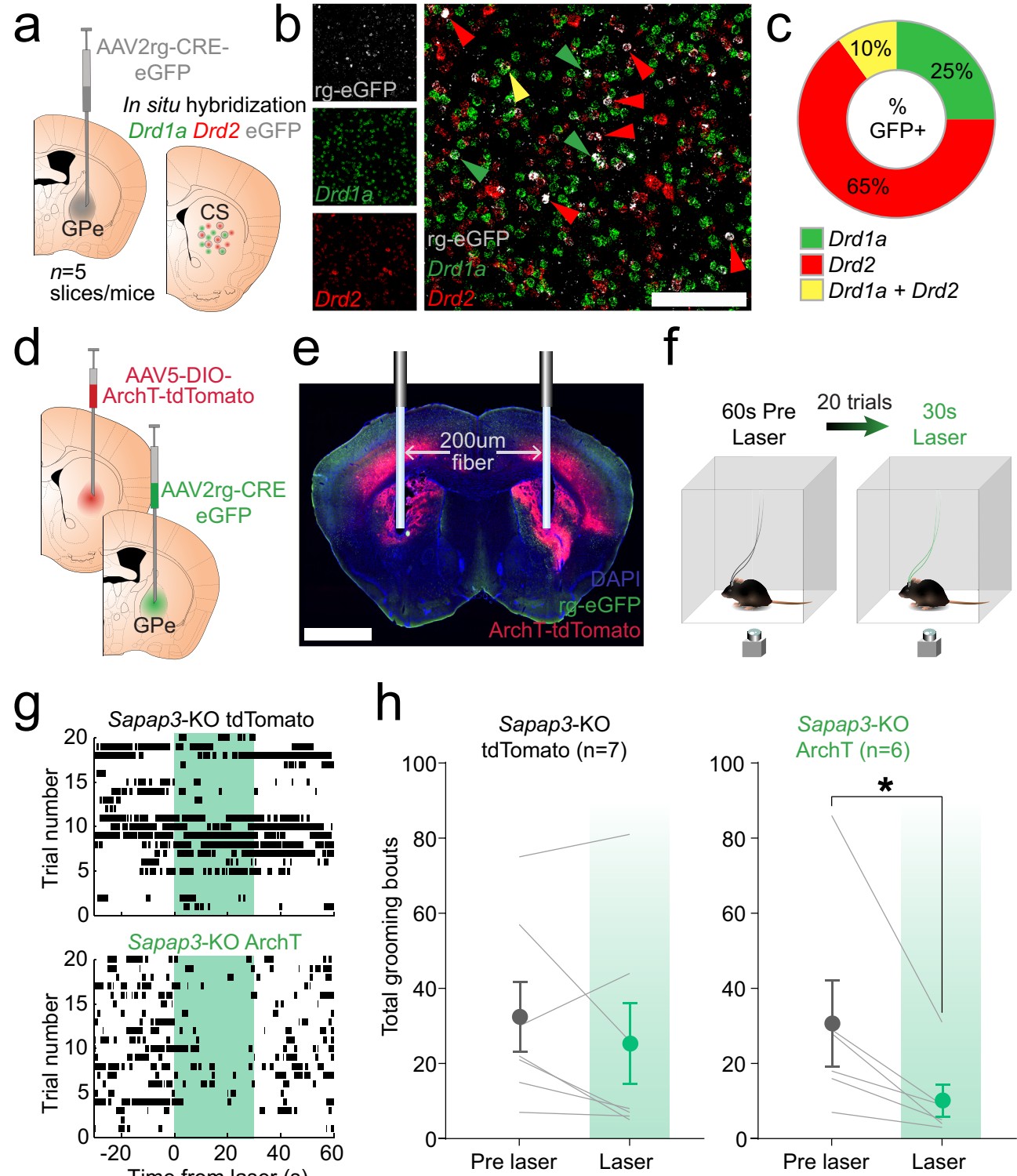

D2- (*Drd2*) SPNs in CS (Fig. 4a). We found that most eGFP+ neurons (65%) expressed only *Drd2*, while a minority (25%) expressed only *Drd1a*, and 10% expressed both markers (Fig. 4b, c; Fig.S7a). Having established this retrograde strategy as an effective way of interrogating the D2-SPN enriched striatopallidal pathway, we determined whether reducing activity of this circuit affects grooming behavior in KOs. We again injected a retrograde AAV2-Cre-eGFP bilaterally into GPe prior to injection of Cre-dependent AAV5-CAG-DIO-ArchT-tdTomato or AAV5-Ef1a-DIO-tdTomato control virus into CS of WT and KO mice (Fig. 4d). Bilateral optic fibers were then placed into CS (Fig. 4e;

Fig.S7b, c). Following viral expression, mice underwent a grooming session during which they received twenty 30 s periods of constant 532 nm laser illumination separated by 60 s periods without light (Fig. 4f). Since grooming is a relatively rare behavior, this paradigm ensured that optogenetic inhibition would coincide with grooming events throughout the experimental session. We hypothesized that by reducing or preventing striatopallidal hyperactivity from occurring, we would thereby reduce the likelihood of excessive grooming bouts occurring in *Sapap3*-KOs. We found that laser illumination did not affect the number of grooming bouts (Fig. 4g, h) or time spent

**Fig. 4 | Inhibition of striatopallidal iSPNs reduces compulsive grooming behavior. a** Schematic of strategy for assessing iSPN expression of dopamine D1 and D2 receptors. Retrograde AAV2-CRE-eGFP was injected into globus pallidus external segment (GPe) and in situ hybridization performed 10 weeks later in CS for eGFP, *Drd1a* and *Drd2*. Brain atlas overlay used with permission of Elsevier Science and Technology Journals from Paxinos and Franklin's the Mouse Brain in Stereotaxic Coordinates, Franklin Keith B.J., Paxinos, George, volume 5, copyright year 2019; permission conveyed through Copyright Clearance Center, Inc. **b** In situ hybridization image for retrograde eGFP (grey), *Drd1a* (green), and *Drd2* (red). Arrows indicate eGFP+ neurons overlapping with *Drd1a* (green), *Drd2* (red), or both *Drd1a* and *Drd2* (yellow). Scale bar = 100 um. **c** Average percentage of eGFP+ neurons expressing *Drd1a* (25%), *Drd2* (65%), or both (10%) in CS (*n* = 4 CS slices from 4 *Sapap3*-WT mice). **d** Schematic for inhibition of iSPNs. Retrograde AAV2-CRE-eGFP was injected bilaterally into GPe and AAV5-CAG-DIO-ArchT-tdTomato or AAV5-DIO-tdTomato was injected bilaterally into CS of WT and KO mice. Brain atlas overlay used with permission of Elsevier Science and Technology Journals from Paxinos and Franklin's the Mouse Brain in Stereotaxic Coordinates, Franklin Keith B.J., Paxinos, George, volume 5, copyright year 2019; permission conveyed through Copyright Clearance Center, Inc. **e** 200 um optic fibers were placed bilaterally into CS; scale bar = 2 mm. **f** Mice received 20 trials of 30 s of 532 nm laser stimulation followed by 60 s of no laser, while behavior was recorded from a bottom view camera. **g** Raster plots of grooming bouts (black tic marks) during laser trials in representative *Sapap3*-KO tdTomato (top) and ArchT (bottom) mice. Green shading indicates duration of laser illumination (30 s). **h** No effect of laser illumination on number of grooming bouts in *Sapap3*-KOs expressing tdTomato control virus (*n* = 7, 2 male/5 female; Two-sided Wilcoxon matched-pairs signed rank test, *p* = 0.23). In *Sapap3*-KO mice expressing ArchT in iSPNs, laser illumination significantly reduced number of grooming bouts (*n* = 6, 2 male/4 female; Two-sided Wilcoxon matched-pairs signed rank test, *p* = 0.031). *$p \leq 0.05$. Green shading indicates laser illumination. Data are presented as mean values +/− SEM. Source data are provided as a Source Data file.

grooming (Fig. S8a) in KOs expressing the tdTomato control virus. In contrast, KOs expressing ArchT displayed a significant reduction in the total number of grooming bouts (Fig. 4g, h) and total amount of time spent grooming (Fig. S8a) during laser illumination (30 s) compared to the period immediately prior to laser illumination (30 s). This effect was observed during laser illumination at two different laser powers (10 mW and 30 mW; Fig. S8c, d), indicating a consistent suppression of indirect pathway activity. Inhibition of iSPNs did not alter locomotion in KOs (Fig. S8b). Finally, inhibition of striatopallidal SPNs did not affect number of grooming bouts or grooming time in WTs (Fig. S8e, f). Together, these data suggest that pathway specific inhibition of primarily D2-SPNs (75%), which are hyperactive and differentially recruited in KOs compared to WTs (Fig. 3), reduces excessive grooming behavior.

## Chronic fluoxetine treatment is associated with a reduction in striatopallidal SPN hyperactivity

SRIs are efficacious for treating OCD in humans[52–55] and reducing compulsive behaviors in mouse models[11,56,57], including compulsive grooming in *Sapap3*-KO mice[33]. However, the mechanisms through which SRIs reduce compulsivity are unclear. We therefore tested whether chronic treatment with the SRI fluoxetine could normalize the neural correlates of compulsive behavior in striatopallidal (primarily *Drd2*-expressing) SPNs–chiefly the increased grooming-associated calcium event rates and increased proportion of grooming-onset activated neurons that were observed in D2-SPNs (Fig. 3). We again used a circuit-based approach by injecting an untagged retrograde AAV2-Cre unilaterally into GPe and a Cre-dependent AAV9-DIO-GCaMP6m into CS ipsilateral to the GPe injection. A GRIN lens was then placed just above the viral injection in CS (Fig. 5a, b; Fig.S9a). At baseline, mice imaged using the striatopallidal circuit-based approach displayed striatal dysfunction similar to that observed when recording D2-SPNs using an A2a-Cre based genetic strategy (Fig. 3), with KO mice having 1) a greater proportion of striatopallidal SPNs activated during excessive grooming (Figs. S9b), and 2) elevated calcium event rates selectively during grooming (Fig. S9c). No significant difference was observed in striatopallidal decoding of grooming behavior (Fig. S9d, e) in *Sapap3*-KOs vs. -WTs. These data provide confidence that striatopallidal SPN dysfunction closely matches what we observe in genetically defined D2-SPNs (Fig. 3) and is well suited to assess potential mechanisms of SRI treatment response using a translationally relevant circuit-based strategy.

After undergoing a 40 min baseline imaging session (Supplementary Movie 4), mice received injections of fluoxetine (5 mg/kg, i.p.) for the next 7 days before undergoing a second grooming and imaging session. Unlike prior reports[33], we did not detect a significant reduction in grooming behavior in KOs following 7 days of fluoxetine treatment (Fig. S10a–c). We therefore continued treatment using a higher dose of fluoxetine (18 mg/kg) via drinking water, which has been shown to achieve blood levels similar to high doses used to treat OCD in people[58], and has been previously used by our group to reduce excessive grooming in *Sapap3*-KOs[14]. Following 4 weeks of treatment, mice underwent another imaging session. Fluoxetine was then removed from all cages for two weeks, followed by a washout imaging session (Fig. 5c). Consistent with prior reports[14], chronic fluoxetine treatment significantly reduced the number of grooming bouts in KOs but did not significantly alter time spent grooming (Fig. 5d). Fluoxetine treatment did not significantly affect grooming bout duration or locomotion in KOs (Fig. S10d, e). Following washout, the number of grooming bouts in KOs returned to baseline levels (Fig. 5d). Fluoxetine did not significantly affect grooming behavior in WTs (Fig. S10g–i).

A reduction in calcium event rates during grooming after fluoxetine treatment was observed when cells were averaged within each mouse, with no changes observed in event rates during non-grooming periods (Fig. 5e). Given that this retrograde approach was selective for only indirect pathway-projecting SPNs and therefore lower yield, we also compared all cells across mice grouped by treatment day. We found a significant reduction in grooming-associated calcium event rate after fluoxetine treatment which returned to baseline after washout (Fig. S10f). These data suggest that fluoxetine-mediated reduction of iSPN activity leads to a reduction in compulsive behavior. The link between compulsive behavior and activity of striatopallidal iSPNs was further solidified by observations that fluoxetine treatment reduced the ability of iSPN activity to predict elevated grooming in *Sapap3*-KOs as measured by F1 score, and predictive precision increased back to baseline levels following washout (Fig. 5g). In addition, we previously identified an increased proportion of activated D2-SPNs during excessive grooming in KOs compared to WT mice (Fig. 3f). Here we find that fluoxetine treatment reduces the percentage of striatopallidal iSPNs activated at the onset of grooming in *Sapap3*-KOs without affecting percentages of grooming-onset inhibited neurons (Fig. 5i–j). A significant positive correlation was observed between the percent change in grooming bouts and the percent change in activated iSPNs following fluoxetine treatment (Fig. 5k), supporting that fluoxetine effects on iSPN activity are involved in anti-compulsive efficacy. Collectively, these results demonstrate that reducing activation of striatopallidal iSPNs is associated with reduced compulsive behavior and may be a mechanism through which SRIs exert their therapeutic effects in OCD.

## Discussion

Although abnormal striatal activity has been consistently associated with compulsive behavior in both human and animal studies, the cellular contributions to these activity patterns is unknown. Here we demonstrate that elevated grooming behavior in *Sapap3*-KO mice is associated with increased recruitment of CS neurons that are activated

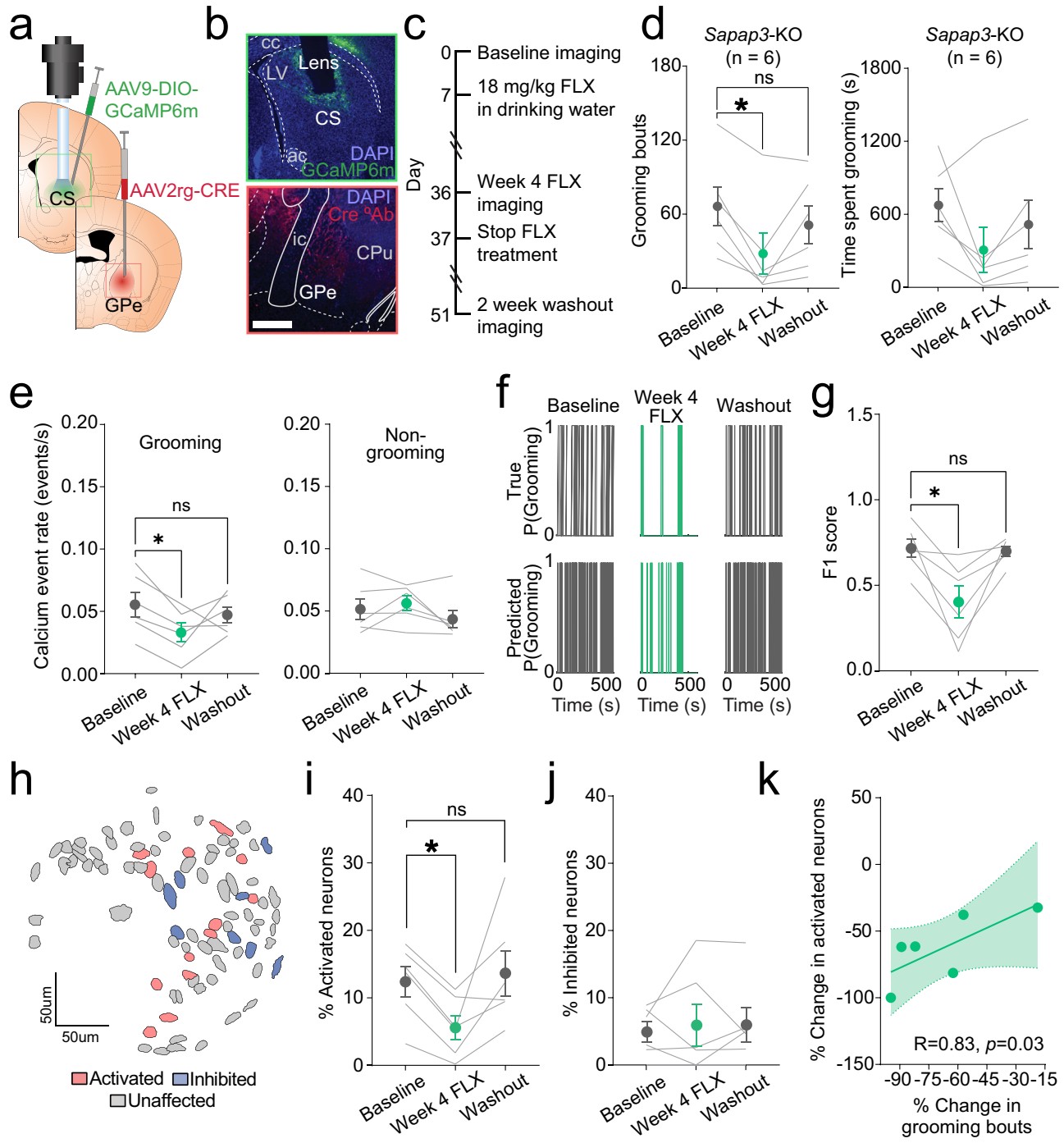

at grooming onset. In addition, population activity of CS neurons more precisely classifies excessive grooming in *Sapap3*-KOs compared to grooming behavior in -WT littermates (Fig. 1). Using a genetic approach to selectively image the two major populations of striatal output neurons, we found that dopamine D1-receptor expressing SPNs did not show grooming-onset hyperactivity in *Sapap3*-KOs, and population activity of these neurons did not predict grooming in -KOs better than -WTs (Fig. 2). Surprisingly, we instead observed grooming-onset hyperactivity of D2-SPNs in KO mice compared to WTs. This hyperactivity stemmed from an increase in the proportion of D2-SPNs that were activated at the onset of grooming. In addition, D2-SPN population activity better predicted grooming in *Sapap3*-KOs relative to -WTs. Using an unsupervised clustering algorithm, we also found that higher proportions of D2-SPNs in KOs were represented in clusters

characterized by heightened activity at onset of grooming and throughout grooming (Fig. 3).

Given that circuit-based therapies for compulsive behavior, such as transcranial magnetic stimulation (TMS) and deep-brain stimulation (DBS), are promising emerging avenues for treatment in severe disease, we next capitalized on a circuit-based approach that would be more amenable to translation into humans[59]. Using a retrograde viral strategy to selectively inhibit indirect pathway-projecting SPNs (which primarily express D2-receptors) (Fig. 4d–f), we found that optogenetic inhibition of striatopallidal SPNs reduced grooming in KO mice (Fig. 4g, h) without affecting grooming in WTs (Fig.S8e, f). Using the same circuit-based approach, we found that chronic treatment with a first line OCD therapeutic, fluoxetine, reduced compulsive behavior, striatopallidal SPN hyperactivity, and recruitment of grooming-onset

**Fig. 5 | Fluoxetine reduces compulsive grooming behavior and iSPN hyperactivity. a** Schematic of strategy for assessing effect of fluoxetine on iSPN activity during compulsive grooming. Retrograde AAV2-CRE was injected unilaterally into GPe and AAV9-DIO-GCaMP6m into ipsilateral CS followed by GRIN lens placement. Brain atlas overlay used with permission of Elsevier Science and Technology Journals from Paxinos and Franklin's the Mouse Brain in Stereotaxic Coordinates, Franklin Keith B.J., Paxinos, George, volume 5, copyright year 2019; permission conveyed through Copyright Clearance Center, Inc. **b** (top) GCaMP6m expression in iSPNs and GRIN lens track in CS. (bottom) Immunohistochemical stain for Cre-recombinase (red) in GPe. Scale bar = 1 mm. **c** Timeline for experiments evaluating effect of fluoxetine on compulsive grooming behavior and iSPN activity. **d** (left) Fluoxetine significantly reduced number of grooming bouts in *Sapap3*-KOs [$n = 6$: 6 female mice; Friedman test ($Fr(3,6) = 7.0$, $p = 0.03$); Dunn's multiple comparisons test, Baseline vs. Week 4 FLX ($Z = 2.6$, $p = 0.03$); Baseline vs. Washout ($Z = 0.86$, $p = 0.99$); Week 4 FLX vs. Washout ($Z = 1.7$, $p = 0.25$)]. (right) No effect of treatment on time spent grooming [Friedman test ($Fr(3,6) = 6.3$, $p = 0.052$)]. **e** (left) Fluoxetine treatment significantly reduced calcium event rates during grooming in KOs [$n = 6$, Friedman test ($Fr(3,6) = 9.0$, $p = 0.008$); Dunn's multiple comparisons test, Baseline vs. Week 4 FLX ($X = 2.6$, $p = 0.02$)]. (right) No effect of fluoxetine on iSPN event rate during non-grooming time [Friedman test ($Fr(3,6) = 4.3$, $p = 0.14$)]. **f** (top) True

grooming behavior during Baseline, Week 4 Fluoxetine, and Washout sessions in representative animal. (bottom) RUSBoost-predicted grooming based on iSPN population activity across days. **g** F1 score for RUSBoost classification of grooming behavior during Baseline, Week 4 Fluoxetine, and Washout sessions [$n = 6$, Friedman test ($Fr(3,7) = 7.0$, $p = 0.03$); Dunn's multiple comparisons test, Baseline vs. Week 4 Fluoxetine, ($X = 2.6$, $p = 0.03$); Week 4 Fluoxetine vs. Washout ($X = 2.6$, $p = 0.03$); Baseline vs. Washout ($X = 0.00$, $p = 0.99$)]. **h** Contour map of striato-pallidal iSPNs from representative KO colored according to baseline activity at grooming onset (red= activated, blue= inhibited, grey= unaffected). **i** Fluoxetine reduces percentage of grooming-onset activated striatopallidal iSPNs [$n = 6$, Friedman test ($Fr(3,6) = 9.33$, $p = 0.006$); Dunn's multiple comparisons test, Baseline vs. Week 4 FLX ($Z = 2.9$, $p = 0.01$); Week 4 FLX vs. Washout ($Z = 2.31$, $p = 0.06$); Baseline vs. Washout ($Z = 0.6$, $p = 0.99$)]. **j** No effect of treatment on percentage of grooming-onset inhibited striatopallidal iSPNs [$n = 6$, Friedman test ($Fr(3,6) = 0.09$, $p = 0.99$)]. **k** Positive correlation between the percent change in grooming bouts and percent change in activated neurons following fluoxetine treatment in KO mice (Spearman $R = 0.83$, $p = 0.03$). **$p \leq 0.01$,*$p \leq 0.05$. cc=corpus callosum, LV=lateral ventricle, ac=anterior commissure, ic=internal capsule, CPu=caudate/putamen. Data are presented as mean values +/− SEM. Source data are provided as a Source Data file.

---

active SPNs (Fig. 5). Furthermore, the precision of striatopallidal SPN population activity to predict grooming was reduced following fluoxetine treatment. Together these results demonstrate that indirect pathway striatopallidal SPN hyperactivity contributes to generation of excessive grooming behavior in this model system and suggest that strategies aimed at reducing such hyperactivity could be beneficial for treating compulsive behaviors in disorders like OCD.

Striatal hyperactivity is associated with compulsive behavior in humans[3–5], and treatments that reduce compulsive behavior typically normalize striatal hyperactivity[10,60]. Similar striatal hyperactivity and treatment response profiles have been observed in preclinical studies using a variety of mouse models that display robust compulsive or perseverative behaviors[11,12,33,57,61]. It has long been theorized that hyperactivity of the basal ganglia direct pathway (or reduced activation of the indirect pathway) might underlie compulsive behaviors, including those observed in OCD[5,6,8,23]. Studies in rodents have provided some support for this type of pathway imbalance in the generation of qualitatively similar aberrant behavioral phenotypes, including dyskinesia[25–29] and compulsive drug seeking[62–64]. Our results suggest that this heuristic is not accurate for compulsive grooming behavior, with hyperactivity instead resulting from increased proportions of grooming-onset activated D2-SPNs and not D1-SPNs in KO mice. One notable difference between compulsive grooming and behaviors such as dyskinesia and compulsive drug seeking (in which a D1-SPN-driven imbalance toward the direct pathway is observed) is that grooming is typically a highly stereotyped, serially ordered behavior[47,48,65]. Critically, compulsive grooming behavior in KO mice contains the same sequences and motor components as grooming in WTs, but the frequency, duration, and organization of these sequences is disturbed[66]. This disturbance has profound ramifications for a mouse's well-being, resulting in facial lesions and disrupted sleep[33] as well as impaired ability to mate and care for pups (unpublished observations, Ahmari laboratory). Thus, similar to compulsive behavior in humans with OCD (e.g. compulsive hand-washing in an individual with a contamination obsession), compulsive grooming behavior in KO mice represents an excessive, repetitive version of a normal behavior that interferes with normal activities[13].

We used an unsupervised clustering analysis to further examine the patterns of neural activity associated with the onset and maintenance of grooming behavior in both genotypes. Heterogenous response profiles were observed across the population in both D1-SPNs and D2-SPNs mirroring previous work[18,19], highlighting the importance of coordinated activity between these populations in the normal control of behavior. Across the 8 clusters identified for D2-

SPNs, KO mice showed increased proportions of clusters comprised of neurons with increased activity at the onset of grooming (Clusters 1 and 3) and late in grooming bouts (Cluster 4), suggesting that striatal hyperactivity during grooming results from these grooming-activated clusters. In contrast, D1-SPNs showed mixed changes in cluster identity across genotypes, with KOs showing decreased proportions of grooming-onset (Cluster 2) and offset (Cluster 6) activated neurons, but increased proportions of some other grooming-onset activated neurons (Cluster 3) as well as neurons that were inactive during grooming (Cluster 8). Taken together with our observations of no change in grooming-onset fluorescence and grooming-associated calcium event rates in D1-SPNs (Fig. 2d-e), these clustering observations suggest that D1-SPNs cannot explain the striatal hyperactivity associated with grooming onset in KOs. However, increases in the proportion of both D1- and D2-SPNs that are active following the onset of grooming bouts (Cluster 3) in KO mice may suggest that coordinated activity of these populations supports some aspect of compulsive grooming; this question can be explored in future studies. Studies examining striatal control of sequential behavior appear to shed light on our finding that D2-SPN hyperactivity promotes compulsive grooming in *Sapap3*-KOs. Optogenetic activation of D2-SPNs (as well as inhibition of D1-SPNs) reduced the ability of wild-type mice to complete sequences on a well-trained serial order task[67], instead leading to an increase in perseverative sequence start responses. This raises the possibility that suppression of D2-SPN activity is necessary for successful completion of action sequences in WT mice. Conversely, D2-SPN hyperactivity might promote compulsive behavior due to excessive grooming sequence initiation combined with the inability to complete serial grooming bouts known as syntactic grooming chains. This interpretation is supported by our and others' data that KO mice engage in many more individual grooming bouts than WT mice[14,33,68]. We expand on these observations by identifying a neural substrate through which reducing D2-SPN hyperactivity might exert its anti-compulsive effects–i.e., decreasing recruitment of functional clusters of grooming-onset activated iSPNs (Fig. 3). Furthermore, we demonstrate that an effective treatment for OCD suppresses activity of iSPNs that preferentially express D2-receptors, providing evidence that this may be an important therapeutic mechanism for the treatment of OCD.

While recently published work supports the idea that inhibiting genetically defined D2-SPNs reduces compulsive grooming behavior in *Sapap3*-KOs[61], this type of manipulation is unlikely to be amenable to translation, since treatment with drugs targeting the D2-receptor produces substantial unwanted side effects[69]. In contrast, our circuit-

based approach to inhibiting striatopallidal SPNs may more readily inform translation into humans in the form of TMS or DBS. Using a retrograde strategy to infect GPe-projecting SPNs, we found that striatopallidal projections from the associative (central) striatum are comprised of 75% *Drd2*-expressing SPNs and 35% *Drd1a*-expressing SPNs (10% express both *Drd2* and *Drd1a*). To our knowledge, this is the first quantitative assessment of striatopallidal SPN identity conducted in the associative striatum, though evidence for pallidal innervation by D1-SPNs has previously been found[70,71]. Our findings are consistent with previous functional work demonstrating innervation of the GPe by a small percentage of D1-SPNs (18%)[72]. Aptly named "bridging" collaterals from D1R-expressing SPNs to the pallidum have the ability to bridge the direct and indirect pathways, modulating basal ganglia output activity by adjusting the balance between the two pathways[70,73–75]. It is unlikely that bridging collaterals are solely responsible for the observed grooming-onset hyperactivity of striatopallidal SPNs in *Sapap3*-KOs, since this was consistent with our finding that D2-SPNs are hyperactive at grooming-onset in *Sapap3*-KO mice (Fig. 3) and the majority of identified neurons in the striatopallidal projection are D2-SPNs (Fig. 4a, b). However, we cannot rule out the possibility fluoxetine treatment directly modulates the balance of direct and indirect pathway activity by regulating the extent of bridging collaterals. While not robustly affected during compulsive grooming, hierarchical clustering did reveal differences in proportions of several D1-SPN functional clusters (Fig. 2i, j), which could contain D1R-expressing SPNs with bridging collaterals. This is an important area of investigation to be explored in future work.

Together with previous findings, our work also supports a dorsal/ventral gradient in pallidal innervation by striatal D1- and D2-SPNs, with roughly 50% of ventral striatal D1-SPNs innervating ventral pallidum[72], while only ~25% of central striatal D1-SPNs innervate the GPe (Fig. 4a–c). Our findings are also in agreement with previous work in the ventral striatum indicating that innervation of the pallidum by D1-SPNs cannot simply be accounted for by SPNs that express both *Drd2* and *Drd1a*[72]. Importantly, our demonstration that a high proportion (25%) of striatopallidal SPNs in the central striatum do not express D2-receptors emphasizes the need to explore pathway specific manipulations rather than solely genetic strategies to better understand how compulsive behavior is generated by distinct striatal output pathways.

A key question raised by our experiments is what drives striatopallidal, D2-SPN hyperactivity in *Sapap3*-KO mice? In wildtype mice, activation of cortical inputs to the striatum preferentially increases immediate-early gene expression in indirect pathway-projecting striatal neurons[76,77]. Numerous studies demonstrate that the associative and sensorimotor striatum of KO mice receive altered input from several cortical regions[32,33,36], including the lateral orbitofrontal cortex (lOFC) and supplementary motor region (M2; also known as anterolateral motor cortex; ALM)[12,37,78]. We have shown that M2 connectivity to both central striatal SPNs and fast-spiking interneurons (FSIs), which strongly inhibit nearby SPNs[79–81], is significantly increased in KO mice relative to WTs[37]. Activating this projection promotes naturalistic grooming sequences in WT mice and elicits activation of central striatal SPNs on the same time scale[82], though the precise striatal cell types affected in vivo are not known. Given that striatal FSIs in the dorsal striatum more strongly inhibit D1-SPNs over D2-SPNs[83], it is plausible that increased drive to FSIs in KO mice may lead to imbalanced output whereby greater proportions of striatonigral SPNs (majority D1-expressing) are strongly inhibited compared to striatopallidal SPNs (majority D2-expressing). Here we demonstrate that enhancing inhibition of striatopallidal SPNs via direct optogenetic inhibition or reducing the proportion of grooming-onset activated striatopallidal SPNs via chronic fluoxetine treatment reduces compulsive behavior, perhaps by compensating for decreased FSI-mediated D2-SPN inhibition (or enhanced FSI-mediated D1-SPN inhibition) in KO mice. Of note, the optogenetic approach used here is

independent of ongoing behavior. Previous work suggests that some level of CS neuron activation is necessary to initiate a compulsive grooming bout[12], and our data indicate that both D2- (Fig. 3) and striatopallidal-SPN hyperactivity (Fig. 5) occur at the onset of a compulsive grooming bout, indicating that triggering optogenetic inhibition immediately following grooming onset may be insufficient to stop the behavior once it has already begun. With the evolution of new technologies allowing automated closed-loop optogenetic stimulation, it will be important to determine the temporal specificity of iSPN inhibition causing a decrease in *Sapap3*-KO grooming.

Decreased iSPN activity following fluoxetine treatment may be mediated by direct action in the striatum or indirectly through modulation of cortical circuits that preferentially recruit populations of direct vs. indirect pathway SPNs. It is likely that our findings are driven at least in part by direct striatal modulation, as serotonin increases the excitability of striatal FSIs[84]. This enhanced FSI activity serves to increase the inhibitory tone onto iSPNs, decreasing their overall activity. Direct optogenetic activation of striatal FSIs has recently been shown to reduce grooming in *Sapap3*-KO mice[85], lending further support to this proposed mechanism of fluoxetine actions in this model. Serotonergic modulation of FSIs is mediated by activation of 5HT2c receptors, as application of RS102221 (a selective 5HT2c receptor blocker) reverses serotonin-induced FSI depolarization[84]. Fluoxetine may also directly modulate the balance of direct and indirect pathway activity, as chronic fluoxetine treatment induces ΔFosB expression in D1-SPNs, but not D2-SPNs, in ventral striatum[86]. Interestingly, this ΔFosB induction in ventral striatum is necessary and sufficient for fluoxetine's antidepressant effect[87], and it is plausible that a similar increase in central striatal D1-SPN activity may serve to normalize striatal outflow in *Sapap3*-KOs following chronic FLX treatment. More direct pharmacological manipulations of D2-receptors, including treatment with the partial agonist aripiprazole, have shown some efficacy for augmenting SRI therapy in some treatment refractory OCD subjects[88]. Additionally, aripiprazole was recently shown to reduce primarily short-duration grooming bouts (but not syntactic grooming sequences) and tic-like behaviors in *Sapap3*-KO mice[66]. Future work is needed to determine the precise circuit and molecular mechanisms by which SRIs reduce the recruitment of striatopallidal SPNs.

In conclusion, our work experimentally demonstrates the surprising finding that, contrary to long-held hypotheses, compulsive behavior is driven by hyperactivity of *Drd2*-expressing, striatopallidal SPNs. Furthermore, we show that the first-line pharmacotherapy for treatment of OCD, fluoxetine, reduces striatopallidal SPN hyperactivity, providing a potential mechanism for the treatment of compulsive behavior.

## Methods

### Animals

All procedures were carried out in accordance with the guidelines for the care and use of laboratory animals from the NIH and with approval from the University of Pittsburgh Institutional Animal Care and Use Committee (IACUC). Mice were housed in ventilated cages on a humidity and temperature controlled rack with *ad libitum* access to food and water in a room on a 12:12 h dark:light cycle (lights on at 7:00AM). *Sapap3*-knockout (*Sapap3*-KO) and wildtype (WT) littermates expressing the Cre-recombinase (Cre) transgene were generated by breeding *Sapap3* heterozygous mutants (*Sapap3*+/−) with heterozygous D1-Cre mice (D1-Cre+/−) or heterozygous A2a-Cre mice (A2a-Cre+/−)[89]. Subsequent *Sapap3*+/−::Cre+/− (either D1-Cre+/− or A2a-Cre+/−) mice were mated with a *Sapap3*+/− mouse, allowing for the generation of Cre positive *Sapap3*-WT mice (*Sapap3*+/+::Cre+/−) and Cre positive *Sapap3*-KO mice (*Sapap3*−/−::Cre+/−). Mice were maintained on a 100% C57BL/6 background. Male and female *Sapap3*-KO and wildtype littermates were used (see figure legends for numbers used in each

experiment). Mice in all cohorts were approximately 4–8 months old at the time of initial surgery.

## General stereotactic surgical methods

For all surgeries, mice were anesthetized using 5% isoflurane mixed with oxygen and maintained on 1–2% isoflurane for the duration of surgery on a small-animal stereotactic instrument (Kopf Instruments). All measurements were made relative to an interpolated bregma. Viral injections were performed using a fixed needle Hamilton syringe (Cole-Parmer Scientific, Vernon Hills IL, USA) connected to sterile polyethylene tubing affixed to a metal cannula and a Harvard Apparatus 11 Elite Syringe Pump (Harvard Apparatus, Holliston MA, USA). Stability of the implant was ensured by placement of one or two 0.45 mm skull screws secured just in front of the lambdoid suture. Following completion of each surgery, mice were injected with subcutaneous (s.c.) carprofen (10% w/v in 0.9% saline) and administered topical antibiotic ointment (TAO) and lidocaine around the headcap. For all surgical procedures mice were kept group housed with littermates unless conspecific fighting was noted, in which case the aggressor was isolated for the duration of the experiment.

## In vivo microendoscopy surgical methods

For experiments recording from all central striatal (CS) neuron types, we injected 800 nl of AAV9-Syn-GCaMP6m-WPRE-SV40 (titer $1.98 \times 10^{12}$, Addgene 100838-AAV9, lot v28555) into CS (AP: +0.65, ML: −1.8, DV: −2.9 & −3.0). For experiments in CS in D1-Cre / *Sapap3* mice and A2a-Cre / *Sapap3* mice, 800 nl of a virus encoding Cre-dependent GCaMP6m (AAV9-Syn-Flex-GCaMP6m-WPRE-SV40, titer $2.72 \times 10^{12}$) was injected into CS (AP: +0.65, ML: −1.8, DV: −2.9 & −3.0). To selectively image striatopallidal neurons in CS, 400 nl of a Cre-expressing virus that is retrogradely transported back one synapse (AAV-pmSyn1-EBFP-Cre, titer $1.00 \times 10^{13}$, Addgene 51507-AAVrg, lot 27021) was injected into external segment of globus pallidus (GPe) (AP: +0.15, ML: −2.00, DV: −3.80) and 800 nl of Cre-dependent GCaMP6m (AAV9-Syn-Flex-GCaMP6m-WPRE-SV40, titer $2.7 \times 10^{12}$) was injected into CS (AP: +0.75, ML: −1.80, DV: −2.9 & −3.1). CS injections were done in a two-step manner, with 400 nl of virus injected at DV −2.9 and 400 nl injected at DV −3.0. For all imaging experiments, a 500 μm diameter, 6 mm length gradient refractive index lens (ProView GRIN lens, Inscopix Palo Alto, CA USA) was lowered just dorsal to viral injection target (AP: +0.65, ML: −1.8, DV: −2.85) (DV: −2.80 in retroAAV GPe experiment) for visualization of cells in the target region. For all GRIN lens implants, acrylic dental cement (Lang Dental, Wheeling IL, USA) was used to secure the lens in place and seal the entire lens during virus incubation period (4-6 weeks).

After 4-6 weeks for viral incubation, mice were again anesthetized with isoflurane and secured to a stereotactic apparatus using ear cuffs. Using a Dremel, excess dental cement was carefully removed to expose ProView GRIN lens. A magnetic microscope baseplate (Part ID:1050-002192, Inscopix) was then attached to the miniaturized microscope (nVistaHD 2.0 epifluorescence microscope, Inscopix) and lowered into place above GRIN lens with 475 nm blue LED gain and power set to maximum. Optimal field of view was determined by focusing on visible cells or other gross landmarks (blood vessels). Baseplate was then cemented in place and plastic Microscope Baseplate Cover (Part ID:1050-002193; Inscopix) was attached.

## Optogenetic inhibition of striatopallidal SPNs surgical methods

A dual-virus retrograde transfection technique was used to target opsin expression selectively in CS striatopallidal SPNs. 350 nl of retrograde Cre (AAVrg-hSyn-Cre-EGFP, titer $1.3 \times 10^{13}$, Addgene 105540-AAVrg, lot v38984) was injected bilaterally into GPe (AP: −0.3, ML: −2.00/ +2.00, DV: −3.80) and 400 nl of Cre-dependent archaerhodopsin (400 nl pAAV5-CAG-DIO-ArchT-tdTomato, titer $1.30 \times 10^{13}$, Addgene 28305, lot v24870) or tdTomato (AAV5-Ef1a-DIO-tdTomato,

titer $7.3 \times 10^{12}$, UNC viral vector core) were injected bilaterally into CS (AP: +0.75, ML: −1.80/ +1.80, DV: −2.9 & −3.1) of experimental and control groups, respectively. Optic fibers (200um diameter, 0.39 NA) were implanted bilaterally in CS (AP: +0.75, ML: −1.80/ +1.80, DV: −3.00).

## Drug preparation and administration

(±)Fluoxetine hydrochloride (fluoxetine) was obtained through NIMH Chemical Synthesis and Drug Supply Program. Fluoxetine was first administered (5 mg/kg in saline i.p.) once daily for 7 consecutive days as previously described[33]; however, this did not significantly impact excessive grooming in *Sapap3*-KOs (Fig.S10a-c). In the same animals, fluoxetine was next administered via drinking water at higher dose according to established methods[11]. Briefly, bottles were placed in each cage and drinking was monitored for 3 consecutive days. Based on average daily consumption and average weight of mice in each cage, 18 mg/kg of fluoxetine hydrochloride was mixed with autoclaved drinking water and stored in black bottles to prevent degradation. Drinking was monitored daily, and amounts were adjusted as needed to maintain average dose of 18 mg/kg fluoxetine per cage. Each cage was given 100 mg/L of drinking water. Bottles were changed every 4 days to prevent degradation of the fluoxetine. Body weight was monitored 3x per week. After 4 weeks of administration, fluoxetine was removed from cages and mice were given a two-week washout to monitor whether grooming behavior and neural activity differences returned to baseline levels.

## Behavioral apparatus and assessment of grooming behavior and locomotion

A custom-built behavioral apparatus was constructed for delivery of optogenetic stimuli and accurate simultaneous assessment of spontaneous (e.g. grooming) behavior and neural activity via in vivo calcium imaging. A clear plexiglass sheet was suspended over a behavioral acquisition camera (Point Grey Blackfly, FLIR Integrated Imaging Solutions). A clear acrylic chamber (8″ x 8″ x 12″) was placed above the camera. For calcium imaging, behavioral acquisition was conducted at 40 Hz using SpinView (Point Grey) software and detailed frame information was sent directly to a central data acquisition box (LabJack U3-LV, Labjack Corporation, Lakewood CO USA). A randomly flashing (30 s ITI) LED visible in the behavioral video (controlled by custom scripts via an Arduino (Arduino Leonardo, Somerville MA, USA)) and sending TTL pulses to the LabJack was used for alignment of behavior and calcium data. For optogenetics, behavioral acquisition was conducted at 40 Hz using SpinView software, and an Arduino controlling laser stimulation also controlled a visible LED in the field of view to synchronize optogenetic stimulation periods to behavior.

Following acquisition, video was converted and compressed (maintaining accurate frame rate information) into .MP4 format using the open-source software HandBrake. Videos were then imported into Noldus The Observer XT (Noldus, Leesburg VA, USA) and grooming behavior was scored frame by frame. Grooming behavior was scored according to previous reports[14,65] by an observer blind to experimental condition (genotype and drug treatment). A mouse was considered to be grooming if it engaged in any of the following behaviors: 1) Facial grooming: touches its face, whiskers, or head with its forepaws; 2) Body grooming: licks its flank or its ventral surface; 3) Hind leg scratching: uses one of its hind legs to scratch its flank, neck or head. These three subtypes comprise the vast majority of mouse spontaneous grooming[65,90]. The beginning of a grooming bout was defined as the frame when a mouse made a movement to begin grooming (e.g., a face grooming bout began in the frame that a mouse lifted its paw off the ground to touch its face). The end of a grooming bout was defined as the frame when a mouse ceased grooming (e.g., a body grooming bout ended when the mouse moved its snout from its flank). Consecutive grooming bouts separated by less than one second were

collapsed into the previous bout. Thus, the minimum amount of time possible between grooming bouts for all experiments was one second.

For locomotion assessment, behavior videos were imported into Noldus EthoVision XT (Noldus, Leesburg VA, USA). 40 Hz video files were used for analysis. A square arena was drawn encompassing the entire behavioral chamber filmed from below. Gray-scaling was used to automatically detect a mouse's location within the arena. Within a single experiment, gray-scaling values remained constant across mice. Distance (centimeters), velocity (cm/s), and X/Y location in space was tracked. For imaging experiments, movement/locomotion onset was identified when a mouse's acceleration exceeded 2 cm/s. A 5 frame averaging window was used to minimize outliers.

## In vivo calcium imaging in freely moving mice

After at least 1 week recovery from baseplate surgery, mice were habituated to attachment of the microscope. Mice were lightly scruffed and the miniature nVistaHD 2.0 microscope was connected to the magnetic baseplate and secured with a set screw. During habituation, optimal focus, field of view, and LED power and gain settings were determined visually by assessing presence of clearly defined putative neurons. A caliper was used to measure the microscope focus to ensure multiple imaging sessions were conducted with same field of view. Mice were also habituated via placement into the acrylic chamber under low light conditions for 5–10 min daily for three days prior to imaging.

Following habituation, mice were given a 40 min baseline behavior and imaging session. Under low light, the microscope was attached, and mice were placed into a temporary holding cage. Mice were given 3–5 min after scope attachment for recovery from scruffing and to allow any rapid photobleaching to occur. Mice were then carefully placed into clear acrylic chamber. LabJack data acquisition then began, immediately followed by behavioral SpinView recordings and nVistaHD software recording compressed greyscale tiff images at 20 Hz. As with behavioral frame acquisition, individual calcium frame information was sent to LabJack for subsequent alignment of behavior and calcium data. Analog gain of image sensor was set between 1 and 4 and 470 nm LED power was set between 10-60% transmission range. For retrograde-Cre CS imaging, settings were kept consistent for each mouse throughout all fluoxetine and washout imaging sessions.

## Optogenetic inhibition of striatopallidal SPNs during spontaneous behavior

Mice were habituated to attachment of optical patch cables in the behavioral chamber under low light conditions for 5 consecutive days prior to experiment (5 min). The interface between patch cable and fiberoptic ferrule was covered in black heatshrink during the experiment to minimize light leak. Mice were connected to patch cables and placed in experimental chambers. Mice were given 3–5 min between connecting patch-cords and beginning the optical inhibition protocol. Mice first underwent one session in which they received twenty 30 s periods of continuous 10 mW stimulation from a green 532 nm fiber-coupled high-power laser (Opto Engine, Midvale Utah USA) separated by 60 s periods without light stimulation. One month later, they underwent the same experimental paradigm with 30 mW of laser stimulation.

## Histological confirmation of virus, fiber, and GRIN lens probe targeting

Mice were transcardially perfused with 4% paraformaldehyde (PFA) and phosphate buffered saline (1X PBS). Immediately after perfusion, heads (with implants intact) were placed into 4% PFA for 24 h for post-fixation after which brains were removed and transferred to a 30% sucrose (in 1X PBS) solution. Brains were frozen and 35 μm sections were cut on a cryostat in 1:6 series. 1 series was mounted, counter-stained with DAPI mounting media (Fluoroshield, Sigma-Aldrich) and

imaged at 10x resolution on slide scanning microscope (Olympus VS120) to visualize virus and implant. Lens confirmation was unable to be determined for 9 mice from our experiment recording all central striatal neurons using the hsyn-GCaMP6m virus (Fig. 1, Fig. S1a) due to missing tissue. For experiments imaging striatopallidal neurons using AAVretro-pmSyn1-EBFP-Cre, a standard immunohistochemical protocol was used to enhance weak native fluorescence [mouse α-Cre recombinase primary antibody (1:1500, MAB3120, Millipore); biotiny-lated anti-mouse secondary antibody (1:250 Jackson ImmunoResearch, RRID: AB_2338565); tertiary streptavidin conjugated Cy-3 (1:250 Jackson ImmunoResearch, RRID: AB_2337244)]. Determination of precise virus, fiber, and GRIN lens location was made by overlaying histological images on the Paxinos and Franklin reference atlas[91].

In situ hybridization of striatal SPNs projecting to GPe: RNAscope was performed on fresh frozen striatal slices taken from *Sapap3*-WTs (n = 4) injected bilaterally in GPe with a retrogradely transported Cre-expressing AAV (350 nl AAVrg-hsyn-Cre-eGFP, titer 1.3 × 10^13, Addgene 105540) (AP: −0.3, ML: ±1.98, DV: −3.8). Tissue was collected 10 weeks after viral injection to allow time for retrograde transport and viral expression. 18 μm coronal sections were cut and mounted to slides. In situ hybridization was performed according to RNAscope Multiplex Fluorescent Reagent Kit v2 protocol (Catalog Number 320393) using the following probes: D2 (RNAscope® Probe-Mm-Drd2-C3, catalog #406501), D1 (RNAscope® Probe-Mm-Drd1a, catalog #406491), and eGFP (RNAscope® Probe-Mm-EGFP-C2, catalog #400289). Images were acquired using Olympus Fluoview 3000 con-focal microscope (20x magnification). Two blinded experimenters were provided with images of CS sections (300um x 300um region) in which *Drd1a* and *Drd2* were randomly pseudocolored for different experimental subjects, such that experimenters did not know which color corresponded to which probe. Cells projecting from CS to GPe were labeled with the EGFP probe. Experimenters individually counted the number of individual EGFP+ cells co-labeled with *Drd1a*, *Drd2*, or both, for each sample.

## Calcium imaging data processing

All imaging pre-processing was performed using Mosaic software (version 1.2.0, Inscopix) via custom Matlab (MATHWORKS, Natick MA, USA) scripts. Videos were spatially downsampled by a binning factor of 4 (16x spatial downsample) and temporally downsampled by a binning factor of 2 (down to 10 frames per second). Lateral brain motion was corrected using the registration engine TurboReg[92], which uses a single reference frame to match the XY positions of each frame throughout the video. Motion corrected 10 Hz video of raw calcium activity was then saved as a .TIFF and used for cell segmentation.

Using custom Matlab scripts, the motion corrected .TIFF video was then processed using the Constrained Non-negative Matrix Factorization approach (CNMFe), which has been optimized to isolate signals from individual putative neurons from microendoscopic imaging[39]. Putative neurons were identified and manually sorted by an observer blind to genotype according to previously established criteria and as per our published work[14,40]. For all analyses on fluorescence traces, the non-denoised temporal traces (referred to as the "raw" trace in CNMFe) without any deconvolution were used. For each individual cell, the raw fluorescence trace was Z-scored to the average fluorescence and standard deviation of that same trace. Thus, fluorescence units presented here are referred to as "Z-scored fluorescence" unless otherwise noted. For analyses of calcium events, calcium events were directly extracted from CNMFe (neuron.S) and binarized.

## Calcium imaging and behavior alignment

Custom Matlab (MATHWORKS) scripts were used to conduct analysis of grooming-related calcium activity. Grooming behavior (state events) was exported as timestamps (grooming start and grooming stop), defined as described above, and aligned to calcium time by

recording 5 consecutive pulses of the randomly flashing LED (point events). The offset of Noldus behavior time to nVista calcium time was then subtracted off, leaving the same number of frames for both the behavior and calcium fluorescence. Grooming timestamps were then transferred to a binary/continuous trace of the same length and sampling rate (10 Hz) as each calcium trace via logical indexing (grooming = 1, not-grooming = 0). Timestamps for behavior are converted to the closest matching frame in the calcium recording (maximum error of one frame or ± 100 ms at 10 Hz). Calcium activity could then be aligned to the start and end of a grooming bout.

### Event related activity classification

In order to perform unbiased classification of an individual cell's responsiveness (activated, inhibited, or unaffected) to a behavioral event (e.g. grooming or locomotion onset) we adapted a previously used strategy[42]. For each individual cell, raw calcium traces 10 s prior to grooming onset and 10 seconds after grooming onset (200 total samples at 10 Hz, 100 ms per sample) were shuffled in time for each sample (200x), removing any temporal information that was previously in each trace but maintaining the variance within each grooming bout. This shuffle was then performed 1000 times per cell to obtain a null distribution of grooming associated calcium activity. A cell was considered responsive to grooming onset if its average behavioral event Z-normalized calcium fluorescence amplitude between −0.5 s before grooming onset to 3 s after grooming onset exceeded a 1 standard deviation threshold from the null distribution.

### Grooming classification based on CS population activity

Using custom MATLAB scripts, a binary RUSBoost classifier was trained using the Z-scored fluorescence signal for the entire session from each neuron within a single mouse. Fluorescence signal (predictors) from each neuron was partitioned (*cvpartition*) such that the classifier was trained on 75% of the data and tested on the remaining 25%. A RUSBoost classifier (*fitcensemble*, and 'Method' 'RUSBoost') was then trained on these data for each mouse. The response variable was a mouse's binary grooming vector. Separate models were trained in which the calcium fluorescence for each neuron was randomly shuffled (*randperm*) with no repeats. Given that grooming is a relatively infrequent behavior, comparisons of classifier performance across groups used F1 score $F1 = 2 * \frac{Precision*Recall}{Precision + Recall}$ which is more robust to class imbalance.

### Spectral cluster analysis of grooming-related activity

For spectral clustering, each individual cell's activity across all grooming bouts (−3s before grooming onset, +10 s after grooming onset, 130 total samples at 10hz) was averaged within an individual session. Grooming-onset aligned calcium fluorescence was then averaged across all cells within a single genotype. These grooming-onset aligned calcium traces were then concatenated across both genotype (WT and KO) and cell type (all SPNs, D1-SPNs, and D2-SPNs). Spectral clustering on this combined vector was performed according to previously established methods[49]. Briefly, dimensionality was reduced using principal component analysis (PCA). The number of principal components kept was determined using the standard bend/elbow method in the plot of explained variance (Fig. S4a). Based on this metric, 7 principal components were kept for analysis. Data were then projected into the lower dimensional subspace formed by the principal components and then input into the clustering algorithm[49]. Clustering was performed using the Scikit-learn function *sklearn.cluster.Spectralclustering* with the affinity matrix calculated using a *k*-nearest neighbor connectivity matrix. The optimal number of clusters was determined by maximizing the silhouette score over a grid search over parameters. Stability of clustering was estimated by subsampling various fractions of trials and calculating the adjusted Rand index. These analyses demonstrated that 8 distinct functional clusters existed in our data. Following clustering, the cluster identity of each individual cell was then mapped back on to its genotype (WT or KO), cell-type (all SPNs, D1-SPNs, and D2-SPNs), and mouse for subsequent analysis. Inferences about grooming phase associated with cluster identity are subject to the variability of manual grooming assessment and are based off of trial averages, which combine multiple grooming bout durations.

### Statistical analysis

All data were tested for normality prior to statistical analysis using the D'Agostino-Pearson test. For all tests a corrected alpha was set to 0.05. Analyses were conducted using GraphPad Prism (version 8.0 or 9.0, GraphPad Software, San Diego CA, USA) or MATLAB (Mathworks). Unless otherwise noted, plotted data represent mean ± standard error of measurement (SEM).

### Grooming behavior analysis

In the event of normally distributed data, grooming behavior was analyzed using two-tailed independent samples *t*-tests to compare genotypes and optogenetic manipulations. If sample sizes were too low to run the D'Agostino-Pearson test, or if data were significantly non-normal, nonparametric Mann-Whitney tests were conducted for unpaired comparisons, and the Wilcoxon matched-pairs signed rank test was conducted for paired comparisons. For evaluating the effect of fluoxetine treatment on grooming behavior with data that were determined to be normally distributed, one-way repeated measures analysis of variance (ANOVA) was used. Main effects and interactions are reported, and in the case of significant interactions, post-hoc comparisons were made using Sidak's multiple comparison correction. If data were not normally distributed, Friedman's test was used to detect group differences followed by Dunn's multiple comparisons test.

### Analysis of calcium events and calcium fluorescence

For analyses comparing the effect of genotype on binarized calcium event rates, the average calcium event rate per cell was obtained for all grooming and non-grooming periods. The average calcium event rate was then calculated across all cells in a given mouse during each of these periods to obtain a single calcium event rate value for each animal. Calcium event rates during grooming and non-grooming periods in WT and KO mice were then compared with independent samples *t*-tests (normally distributed) or Mann-Whitney tests (non-normally distributed) unless otherwise stated. In experiments involving fluoxetine administration, a repeated measures one-way ANOVA (normally distributed) or Friedman test (non-normally distributed) was used to analyze calcium event rate changes within each genotype. Sidak's post-hoc correction (normally distributed) or Dunn's multiple comparison tests (non-normally distributed) were used for multiple comparisons. For analysis of grooming-onset aligned calcium fluorescence, each individual cell's activity across all grooming bouts (−3s before grooming onset, +10 s after grooming onset, 130 total samples at 10 Hz) was averaged within an individual session, averaging across all neurons from all mice of a given genotype. Grooming- and locomotion-onset aligned calcium fluorescence was then averaged across all cells within a single genotype. A paired *t*-test was then conducted comparing the fluorescence at each sample between genotypes. Because this analysis was conducted across all cells for a given genotype and not averaged across mice, resulting in increased statistical power, a conservative Bonferonni multiple comparison correction was calculated such that the α-value was = 0.00038 (0.05/130). Thus, a significant difference in calcium fluorescence between WT and KO mice for a given sample must obtain a *p*-value of ≤ 0.00038 (MATLAB).

### Analysis of event-related cell classification

For normally distributed two group comparisons, independent samples *t*-tests were used to compare the proportions of modulated cells between genotypes. For non-normally distributed two group comparisons, Mann-Whitney tests were conducted. To analyze the effect of fluoxetine, the non-parametric Friedman test was used followed by Dunn's multiple comparisons test.

### Analysis of RUSBoost binary classifier performance

Comparisons of F1 score across genotype and shuffled data were conducted using a two-way repeated measures ANOVA. Post-hoc comparisons were made using Sidak's multiple comparisons test.

### Analysis of spectral clustering proportions across genotype

The proportion of neurons in each cluster was compared between genotypes using a Chi-squared test (MATLAB).

### Reporting summary

Further information on research design is available in the Nature Portfolio Reporting Summary linked to this article.

## Data availability

The supporting data generated in this study are provided in the Source Data file. Source data are provided with this paper.

## Code availability

The custom-written MATLAB code used to analyze the data from the current study are available via Zenodo https://doi.org/10.5281/zenodo.10790736.

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

## Acknowledgements

These studies were supported by BRAINS R01MH104255, McKnight Neuroscience Scholar Award, MQ Fellows Award, Burroughs Wellcome Fund CAMS, Klingenstein Fellowship in the Neurosciences, and NIMH R01MH119837 to S.E.A.

## Author contributions

S.C.P., E.E.M., and S.E.A. conceptualized the project. S.C.P., E.E.M., and B.L.C. performed surgeries and conducted calcium imaging, optogenetic manipulation, and pharmacological experiments. B.L.C., Z.L., N.M.B, and J.L.P analyzed and manually scored behavioral experiments. E.E.M. and J.L.P. performed and analyzed in situ hybridization experiments. S.C.P., J.H., and V.M.K.N. wrote code for neural data analysis. S.C.P. conducted data analysis. S.C.P., E.E.M., and B.L.C. jointly generated figures. S.C.P., E.E.M., B.L.C., and S.E.A. wrote the manuscript.

## Competing interests

The authors declare no competing interests.
