## [Peer Review File · Nature Communications]

Hyperactivity of indirect pathway-projecting spiny projection neurons promotes compulsive behaviorEditorial Note: This manuscript has been previously reviewed at another journal that is not operating a transparent peer review scheme. This document only contains reviewer comments and rebuttal letters for versions considered at *Nature Communications*.

REVIEWERS' COMMENTS

Reviewer #1 (Remarks to the Author):

Piantadosi, Manning, and colleagues have done an admirable and complete job of addressing the concerns regarding their manuscript titled "Hyperactivity of indirect pathway-projecting spiny projection neurons drives compulsive behavior", which describes a set of experiments aiming to understand the contribution of striatal pathways to compulsive grooming in a mouse model for OCD. I congratulate the authors on the significant improvement of their work, which now instills more solid confidence in their findings and highlights their innovative approach to understanding the neurobiology of compulsive behavior. Among many improvements, I would like to specifically underline the value of the added data provided in Fig.S9.

Since the authors reasoning is convincing in their rebuttal letter, I suggest to add the following rebuttal points to the manuscript:

- 1) Add the three reasons for 30s-optogenetic stimulation (response #4).
- 2) List the reasons for why the inflation of power due to the high N of cells is acceptable (as opposed to animals; response #6).
- 3) The weakest point of the rebuttal is response #18, where the authors address the concern that direct pathway neurons have collaterals that branch off to the GP (complicating the interpretation that indirect pathway neurons are targeted specifically). While the authors' intention to use circuit-specific manipulations is good, it nonetheless needs to be a convincing circuit-specific manipulation. The rebuttal that direct-pathway neuron collaterals have not been demonstrated specifically for the central striatal region is very weak. This point deserves attention and should be explicitly mentioned in the discussion of the manuscript, since it concerns a principal part of the work.

Reviewer #2 (Remarks to the Author):

The authors have very convincingly addressed all my concerns by providing detailed further analysis, experiment and statistical test clarifications. The new figures and discussion make the findings more robust and reflect a more comprehensive understanding of the phenomenological and mechanistic effects of the different opto and pharmaco modulation observed.

I have no further concerns about this publication and would like to congratulate the authors for their study.

Reviewer #1 (Remarks to the Author):

Piantadosi, Manning, and colleagues have done an admirable and complete job of addressing the concerns regarding their manuscript titled "Hyperactivity of indirect pathway-projecting spiny projection neurons drives compulsive behavior", which describes a set of experiments aiming to understand the contribution of striatal pathways to compulsive grooming in a mouse model for OCD. I congratulate the authors on the significant improvement of their work, which now instills more solid confidence in their findings and highlights their innovative approach to understanding the neurobiology of compulsive behavior. Among many improvements, I would like to specifically underline the value of the added data provided in Fig.S9.

Since the authors reasoning is convincing in their rebuttal letter, I suggest to add the following rebuttal points to the manuscript:

- 1) Add the three reasons for 30s-optogenetic stimulation (response #4).
- 2) List the reasons for why the inflation of power due to the high N of cells is acceptable (as opposed to animals; response #6).
- 3) The weakest point of the rebuttal is response #18, where the authors address the concern that direct pathway neurons have collaterals that branch off to the GP (complicating the interpretation that indirect pathway neurons are targeted specifically). While the authors' intention to use circuit-specific manipulations is good, it nonetheless needs to be a convincing circuit-specific manipulation. The rebuttal that direct-pathway neuron collaterals have not been demonstrated specifically for the central striatal region is very weak. This point deserves attention and should be explicitly mentioned in the discussion of the manuscript, since it concerns a principal part of the work.

We again thank the reviewer for their critical feedback and feel it has resulted in significant improvement to our manuscript. We have added additional discussion of the following points to the manuscript:

Point 1) Added details to lines 202-205.

Point 2) Added details to lines 644-645 and 648.

Point 3) Added thorough discussion of this point to lines 344-354.

Reviewer #2 (Remarks to the Author):

The authors have very convincingly addressed all my concerns by providing detailed further analysis, experiment and statistical test clarifications. The new figures and discussion make the findings more robust and reflect a more comprehensive understanding of the phenomenological and mechanistic effects of the different opto and pharmaco modulation observed.

I have no further concerns about this publication and would like to congratulate the authors for their study.

We greatly appreciate the feedback and thank the reviewer for their careful review of our manuscript.